# An EpCAM/Trop2 mechanostat differentially regulates collective behaviour of human carcinoma cells

Azam Aslemarz [1,2,4], Marie Fagotto-Kaufmann [1,5], Artur Ruppel [3,6], Christine Fagotto-Kaufmann[1], Martial Balland[3], Paul Lasko [2] & François Fagotto [1✉]

## Abstract

EpCAM and its close relative Trop2 are well-known cell surface markers of carcinoma, but their potential role in cancer metastasis remains unclear. They are known, however, to downregulate myosin-dependent contractility, a key parameter involved in adhesion and migration. We investigate here the morphogenetic impact of the high EpCAM and Trop2 levels typically found in epithelial breast cancer cells, using spheroids of MCF7 cells as an in vitro model. Intriguingly, EpCAM depletion stimulated spheroid cohesive spreading, while Trop2 depletion had the opposite effect. Combining cell biological and biophysical approaches, we demonstrate that while EpCAM and Trop2 both contribute to moderate cell contractility, their depletions differentially impact on the process of "wetting" a substrate, here both matrix and neighboring cells, by affecting the balance of cortical tension at cell and tissue interfaces. These distinct phenotypes can be explained by partial enrichment at specific interfaces. Our data are consistent with the EpCAM-Trop2 pair acting as a mechanostat that tunes adhesive and migratory behaviours.

**Keywords** Tumor-Associated Protein; Cell Adhesion; Actomyosin Cytoskeleton; Cell Cortex; Tissue Mechanics
**Subject Categories** Cancer; Cell Adhesion, Polarity & Cytoskeleton

## Introduction

Contractility of the actomyosin cellular cortex is a central determinant of cell and tissue adhesive and migratory properties. Understanding how contractility is controlled at the global and local scales is thus key to a mechanistic explanation of morphogenetic processes, including cancer metastasis. This is a complex question, because myosin-dependent contractility is implicated in multiple processes, that can be concurrent or even antagonistic. A multitude of regulators have been identified, and one important, long-standing question is how they can carry specific functions while all targeting the same molecular process.

In this context, EpCAM and Trop2 are particularly intriguing candidate regulators. EpCAM, also called TACSTD1, is a cell surface protein specifically expressed in epithelial tissues, and overexpressed in most human carcinomas (Went et al, 2004; Rao et al, 2005). It is an important cancer biomarker for diagnostic and therapeutic purposes (Gires and Bauerle, 2010). While the name EpCAM (Epithelial Cell Adhesion Molecule) was given based on its proposed function as a homotypic cell adhesion molecule (Litvinov et al, 1994; Balzar et al, 1999), EpCAM is currently viewed as a regulator of intracellular signaling, impacting on proliferation (Maetzel et al, 2009; Chaves-Perez et al, 2013) as well as on cell behavior. The latter activity relies on its ability to downregulate myosin contractility (Barth et al, 2018; Fagotto, 2020a; Gaston et al, 2021; Maghzal et al, 2013; Salomon et al, 2017), providing an obvious putative link between EpCAM and cancer metastasis that remains to be established. While there is only one EpCAM in fish and amphibians, a second closely related gene, called Trop2 or TACSTD2, has appeared in reptilians, at the root of amniote evolution, as a result of the duplication of the EpCAM gene. In humans, Trop2 is co-expressed with EpCAM in all epithelia, except in the intestine. Human EpCAM and Trop2 have high sequence similarity between themselves, and with fish and amphibian EpCAM. Trop2 has been also linked to cancer, but, again, its actual impact on cell adhesion and migration, and on cancer invasion remains unclear (Fagotto and Aslemarz, 2020; Lenárt et al, 2020; Švec et al, 2022).

EpCAM morphogenetic function is best understood in zebrafish and Xenopus embryos. In both systems, gain and loss-of-function (GOF and LOF) experiments showed that EpCAM acts positively on cell-cell adhesion and motility (Slanchev et al, 2009; Maghzal et al, 2010). In particular, EpCAM is required during gastrulation for dynamic tissue reorganization through cell-cell intercalation (Slanchev et al, 2009; Maghzal et al, 2010). At later stages of development, loss of EpCAM eventually causes severe defects in tissue integrity (Slanchev et al, 2009; Maghzal et al, 2013). We showed that all the EpCAM GOF and LOF embryonic phenotypes

[1]CRBM, University of Montpellier and CNRS, Montpellier 34293, France. [2]Dept. of Biology, McGill University, Montreal, QC H3A1B1, Canada. [3]LIPHY, UMR5588, University of Grenoble, 38400 Grenoble, France. [4]Present address: SGS, Mississauga, ON L5T 1W8, Canada. [5]Present address: Department of Neurobiology, University of Stuttgart, 70569 Stuttgart, Germany. [6]Present address: CRBM, University of Montpellier and CNRS, Montpellier 34293, France. ✉E-mail: francois.fagotto@crbm.cnrs.fr

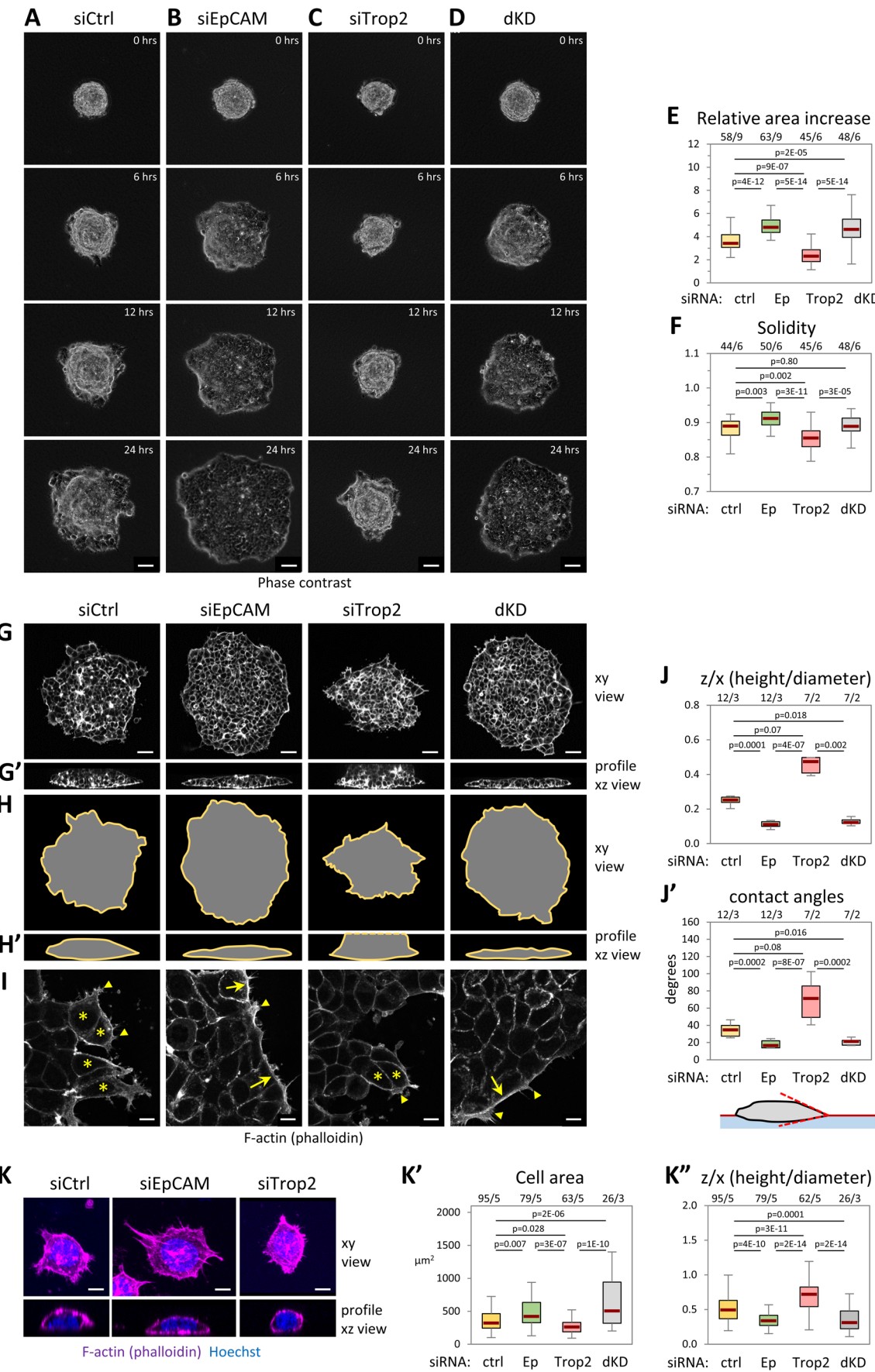

**Figure 1.  Cohesive collective migration of MCF7 spheroids is stimulated by EpCAM KD, but inhibited by Trop2 KD.**

Spheroids of transfected MCF7 cells with control, EpCAM, Trop2, and EpCAM/Trop2 siRNA were plated on a gel of fibrillar collagen, let adhere for 30 min, then phase contrast images were taken every 30 min for 24 h. (A–D) Images of whole spheroids at selected time points. Scale bars: 100 µm. (E) Quantification of relative area increases after 24 h, expressed as the ratio of the final and initial areas. (F) Quantification of spheroids solidity, measured as the ratio [Area]/[Area of convex hull]. The box plots show the interquartile range (box limits), median (center line), and min and max values without outliers (whiskers). Numbers of spheroids/biological replicates are indicated above graphs. Statistical analysis for (E, F): One-way ANOVA followed by Tukey-HSD post hoc test. (G) Representative examples of phalloidin-labeled spheroids after 24 h spreading in top and orthogonal view (G'). Scale bars: 100 µm. (H, H') Horizontal and orthogonal outlines of the same spheroids. (I) Details of spheroid edges. Maximal projection of 3 z planes, 1-µm apart. Protrusions are observed under all conditions (arrowheads). However, control and Trop2 KD spheroids have numerous cells protruding out of the main cell mass (asterisks). These are rare in EpCAM KD and dKD spheroids, which rather display actin cable-like structures along their edge (arrows). Scale bars: 10 µm. (J, J') Morphological parameters: height to diameter ratio (J) and angle at the contact with the matrix (J'). One-way non-parametric ANOVA (Kruskal–Wallis Test) followed by Dunn post hoc test. (K) Effect of EpCAM and Trop2 KD on single-cell morphology. Typical examples of control, EpCAM and Trop2-depleted dissociated cells laid on collagen gel, stained with phalloidin and Hoechst. Top panels show maximal horizontal projection, the lower panel a slice from the orthogonal projection. Scale bars: 10 µm. (K', K″) Quantification of cell area, and calculation of height/diameter ratio. The number of cells/biological replicates is indicated in the above graphs. One-way ANOVA followed by Tukey-HSD post hoc test.

could be accounted for by its ability to downregulate myosin activity (Maghzal et al, 2010, 2013). At the molecular level, EpCAM directly binds and inhibits the novel class of PKC kinases, and by doing so, it represses the phosphorylation of myosin light chain (pMLC) by the PKD-Raf-Erk-MLCK pathway (Maghzal et al, 2013). Importantly, the loss of cadherin adhesion upon EpCAM LOF turned out to be a secondary effect of the acute upregulation of myosin contractility (Maghzal et al, 2013). Myosin repression through nPKC interaction appears conserved in human EpCAM and Trop2 (Maghzal et al, 2013). In human intestinal epithelial cells, where only EpCAM is expressed, its loss in a rare disorder called congenital tufting enteropathy (CTE) leads to disruption of the epithelium (Sivagnanam et al, 2008), due again to myosin overactivation and uncontrolled tension of the actomyosin cortex (Salomon et al, 2017; Barth et al, 2018). Note that a recent study on the migration of isolated intestinal Caco2 cells proposed an additional/alternative role of EpCAM in spatial regulation of RhoA and myosin activity via fast recycling endosomes (Gaston et al, 2021). The significance of this phenomenon in the context of the intact epithelium remains to be evaluated.

While the myosin repression by EpCAM appeared to favor adhesion and migration in the abovementioned models, it is likely to have different effects depending on the cell types and specific conditions, consistent with reported conflicting results on migration of human cell lines (reviewed in (Fagotto and Aslemarz, 2020). Also, while EpCAM and Trop2 are thought to be at least partly redundant (Fagotto and Aslemarz, 2020; Nakato et al, 2020; Szabo et al, 2022; Wu et al, 2020), the conservation of the two genes throughout amniotes argues for distinct functions, although direct experimental comparison at the cellular level has been missing so far. Thus, EpCAM and Trop2 constitute an intriguing case of two very similar myosin regulators, widely co-expressed in epithelia.

In an attempt to clarify their role in a cancer-relevant context, we chose breast epithelial MCF7 cells as a well-established model of EpCAM-positive carcinoma cells (Osta et al, 2004; Dai et al, 2017). MCF7 cells were particularly well suited because EpCAM and Trop2 expression is moderately high, representative of an average range found in breast cancer and other cancer cell lines, and still in the order of magnitude of expression in normal epithelia (Appendix Fig. S1). Furthermore, MCF7 cells display robust E-cadherin-mediated cell-cell adhesion as well as good migratory

activity. EpCAM can be strongly downregulated in particular situations, such as induction of the ectodermal-mesenchymal transition (Sankpal et al, 2017; Vannier et al, 2013; Pan et al, 2018), justifying the physiological relevance of experimental acute manipulations. We thus analyzed the impact of changes in EpCAM and Trop2 levels on MCF7 spheroids spreading on a matrix as a model of dynamic rearrangement of solid tissue, and we systematically characterized the effect on myosin contractility and adhesion, using a combination of cellular and biophysical approaches.

These experimental results were interpreted based on the well-established biophysical analogy with the surface tension of liquids (Steinberg, 1963; Harris, 1976; Ryan et al, 2001; Winklbauer, 2015). Here, cell and tissue spreading and adhesion are controlled by the balance of "tensions" acting at the various interfaces (cell-matrix and cell-cell contacts, and free cell edges), which are largely dominated by cortical actomyosin contractility (Brodland, 2002; Douezan et al, 2011; Maitre et al, 2012; Amack and Manning, 2012; Winklbauer, 2015). This framework has been instrumental as deciphering morphogenetic processes (Manning et al, 2010; Luu et al, 2011; David et al, 2014; Szabo et al, 2016; Canty et al, 2017; Reig et al, 2017; Morita et al, 2017; Shook et al, 2022), and previously used to model the spreading behavior of cell aggregates in vitro (e.g., (Douezan et al, 2011; Gonzalez-Rodriguez et al, 2012; Pérez-González et al, 2019; Efremov et al, 2021; Yousafzai et al, 2022). This model is summarized as a short primer in Fig. EV1.

In this study, we confirmed that EpCAM and Trop2 both inhibit cortical myosin, but we made the unexpected discovery that EpCAM depletion favored adhesion and collective migration, while Trop2 depletion had the converse effect. These opposite phenotypes related to changes in the balance of tension between free and contact cell interfaces, which could be explained by subtle relative enrichments of EpCAM and Trop2, respectively at cell free edges and at adhesive contacts. The analysis of cellular and biophysical parameters highlighted how relatively modest differences in the balance of cellular tensions can deeply impact the morphogenetic outcome. Extension of the study to HCC1500 and MDA-MD-453, two other breast cancer lines with very different characteristics, showed similar differential localization and antagonistic LOF phenotypes, suggesting that these are general properties of the EpCAM/Trop2 pair.

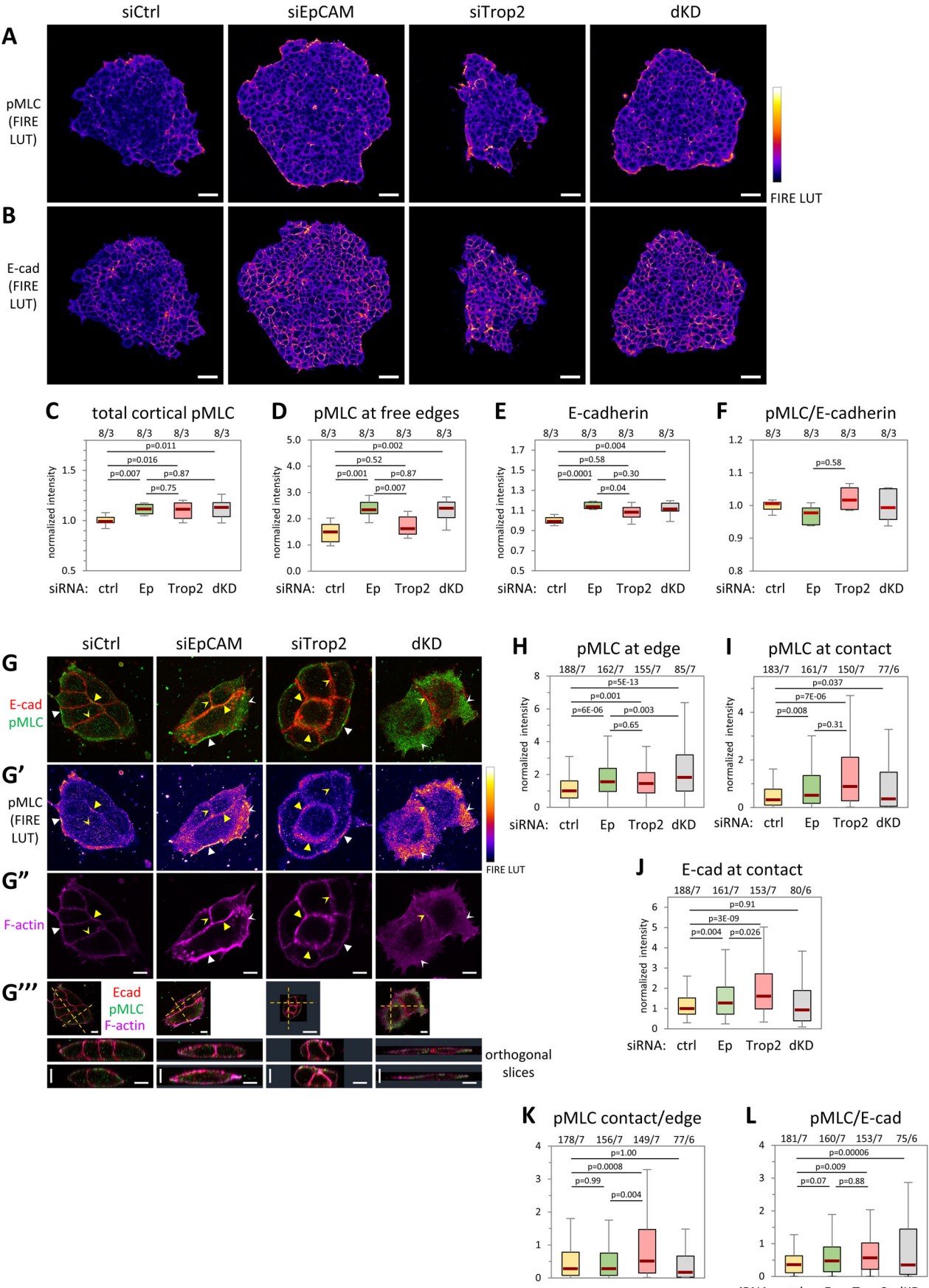

**Figure 2. EpCAM and Trop2 depletions both enhance cortical myosin activation and E-cadherin recruitment, but differentially impact myosin at cell contacts and free edges.**

(A, B) pMLC and E-cadherin levels in spheroids. Representative confocal images of spheroids after 24 h spreading, immunolabelled for pMLC and E-cadherin. Levels are visualized using the "FIRE" pseudocolors (LUT) of ImageJ. The selected z planes correspond to the widest area for each spheroid, at the level of the collagen surface. Scale bar: 50 µm. (C–F) Quantification of normalized mean intensities for total pMLC (C), pMLC along the free edge of the spheroid (D), E-cadherin (E), and calculated pMLC to E-cadherin ratio (F). In all panels, the box plots show the interquartile range (box limits), median (center line), and min and max values without outliers (whiskers). Numbers of spheroids/biological replicates are indicated above graphs. One-way non-parametric ANOVA (Kruskal–Wallis) followed by Dunn post hoc test. (G–L) pMLC and E-cadherin levels in small cell groups. (G) Representative confocal images of small groups of cells immunolabelled for E-cadherin (red) and pMLC (green), F-actin filaments labeled with phalloidin-Alexa647 (A″, magenta). Maximal projection of 5 planes, for a total thickness of 1 µm. (G') pMLC signal shown as FIRE LUT. The image for siTrop2 is a slightly different plane than in (G), to better visualize pMLC at contacts. White-filled arrowheads: Signal along the free edge. Filled yellow arrowheads: Signal at cell contacts. Concave yellow arrowheads: Contacts with low to no pMLC signal. Concave white arrowheads: In very flat cells, the cortical signal is seen from the top (one edge of siEpCAM and the whole dKD pair). (G‴) Comparison of size and shape through profile views (orthogonal slices along the dashed lines). Scale bars, horizontal and vertical: 10 µm. (H–L) Quantification of peak intensities, determined through complete z stack series, for pMLC along free cell edges (H) and cell-cell contacts (I), and E-cadherin (J). (K) Ratios between pMLC contacts and free edges. (L) pMLC to E-cadherin ratios at contacts. pMLC intensity was normalized to the median value at the edges of control cells, E-cadherin with control contacts. Number of cell groups/biological replicates is indicated above graphs. One-way ANOVA followed by Tukey-HSD post hoc test. Source data are available online for this figure.

# Results

## EpCAM depletion increases the spreading of MCF7 spheroids and their cohesiveness, while Trop2 depletion has the opposite effect

In order to investigate the role of EpCAM and Trop2 in collective tissue behavior, we used spheroids of breast cancer MCF7 cells as an in vitro model that mimics a solid tumor (Kramer et al, 2013). These spheroids were placed on a soft matrix of fibrillar collagen I as a physiologically relevant extracellular matrix (Insua-Rodríguez and Oskarsson, 2016). The collagen gel was about 20-50 µm thick and sufficiently soft for spheroids to be capable to deform it (see convex ventral face of spheroids in Fig. 1G',H') (Yousafzai et al, 2022). Under these conditions, spheroids actively spread, which we typically imaged over 24 h (Fig. 1A; Movie EV1). Spreading of a spheroid is an integrated process that involves adhesion to the extracellular matrix and collective migration, together with the ability of the tissue mass to remain coherent while allowing cells to exchange neighbors by cell intercalation. EpCAM was depleted by transfection of siRNA one day before starting to form the spheroids, which were laid on collagen two days later. By that stage, EpCAM depletion was close to complete and remained so until the end of the assay (Fig. EV2A,B). Spheroids formed from cells transfected with control siRNA spread efficiently, typically increasing their area by 3- to 4-fold within 24 h (Fig. 1A,E; Movie EV1). Note that the center of the spheroid tended to remain compact, while the periphery showed irregular contours, with cells or cell groups frequently sticking out of the cell mass (Fig. 1A,G,H,I). EpCAM knockdown (KD) resulted in a strong, highly reproducible increase in spreading (Fig. 1B,E,G; Movie EV2). After 24 h, spheroids had the shape of a flat pancake, only about two to three cells thick (Fig. 1G,H,J). Interestingly, these EpCAM KD spheroids expanded as a very cohesive sheet. While at high magnification, numerous protrusions could be observed that emanated from the cells at the edge of the cell mass, similar to control spheroids (Fig. 1I), at a coarse grain view, this edge was overall strikingly smoother in EpCAM KD (Fig. 1G,H). This distinctive morphology, indicative of higher tissue cohesiveness, was quantitatively expressed as higher "solidity" (Fig. 1F). We validated the specificity of the KD phenotype by showing that it could be fully rescued by using MCF7 cells stably expressing a

DOX-inducible EpCAM variant resistant to the siRNAs (Fig. EV2G). In complementary GOF experiments, EpCAM overexpression decreased spheroid spreading, opposite to the KD phenotype (Fig. EV2H). We also verified that the increased spreading of siEpCAM spheroids was not due to a higher proliferation rate by performing the spheroid assay in the presence of Mitomycin C, an anti-mitotic agent (Fig. EV2F). We concluded that the increased spreading of EpCAM KD was independent of cell proliferation.

Next, we depleted Trop2. Unexpectedly, Trop2 KD alone restricted spheroid spreading, opposite to the effect of EpCAM KD (Fig. 1C,E,G,J; Movie EV3). The shape of Trop2 KD spheroids was strikingly irregular, yielding a low solidity value (Fig. 1E'). Thus, EpCAM and Trop2 appeared to play an antagonistic role in this context, respectively restraining or enhancing both tissue cohesion and spreading. Figure 1E compiles data from two distinct Trop2 siRNAs (sc-43043 and H502-003), which gave very similar results (Fig. EV2C–E). For the rest of the study, we used sc-43043. Transient expression of Trop2-GFP yielded a partial but significant rescue (Fig. EV2I). Using the ectopic expression of EpCAM-GFP and Trop2-GFP to calibrate immunofluorescence (IF) signals, we estimated that the surface levels of EpCAM and Trop2 were quite similar (Appendix Fig. S2). Although Trop2 depletion was not as complete as for EpCAM (~75% versus >95%, Fig. EV2A–D), phenotypes throughout this study were highly reproducible and consistent. Note that depletion of EpCAM KD did not impact the total levels of Trop2, while EpCAM was marginally increased in Trop2 KD (Appendix Fig. S3). We also examined double knockdowns for both EpCAM and Trop2 (dKD) and found that such spheroids did not spread more than EpCAM KD spheroids (Figs. 1D,E,G,J and. EV2E; Movie EV4).

The highly stereotypical morphologies adopted by spreading spheroids upon EpCAM KD and Trop2 KD (Fig. 1G,G',H,H') were suggestive that these molecules acted on the "wetting" properties of the tissue (Fig. EV1). The angle formed between the free edge and the tissue-matrix interface provides a readout for the degree of wetting, strongest for EpCAM KD and lowest for Trop2 KD (Fig. 1J'). Strikingly, we observed the exact same trends for single cells, with EpCAM and Trop2 KD leading respectively to increased or decreased spreading (Fig. 1K,K',K", estimated contact angles in Appendix Fig. S4), which indicated that EpCAM and Trop2 had a similar antagonistic action on the balance of tensions both at the cell and tissue scales.

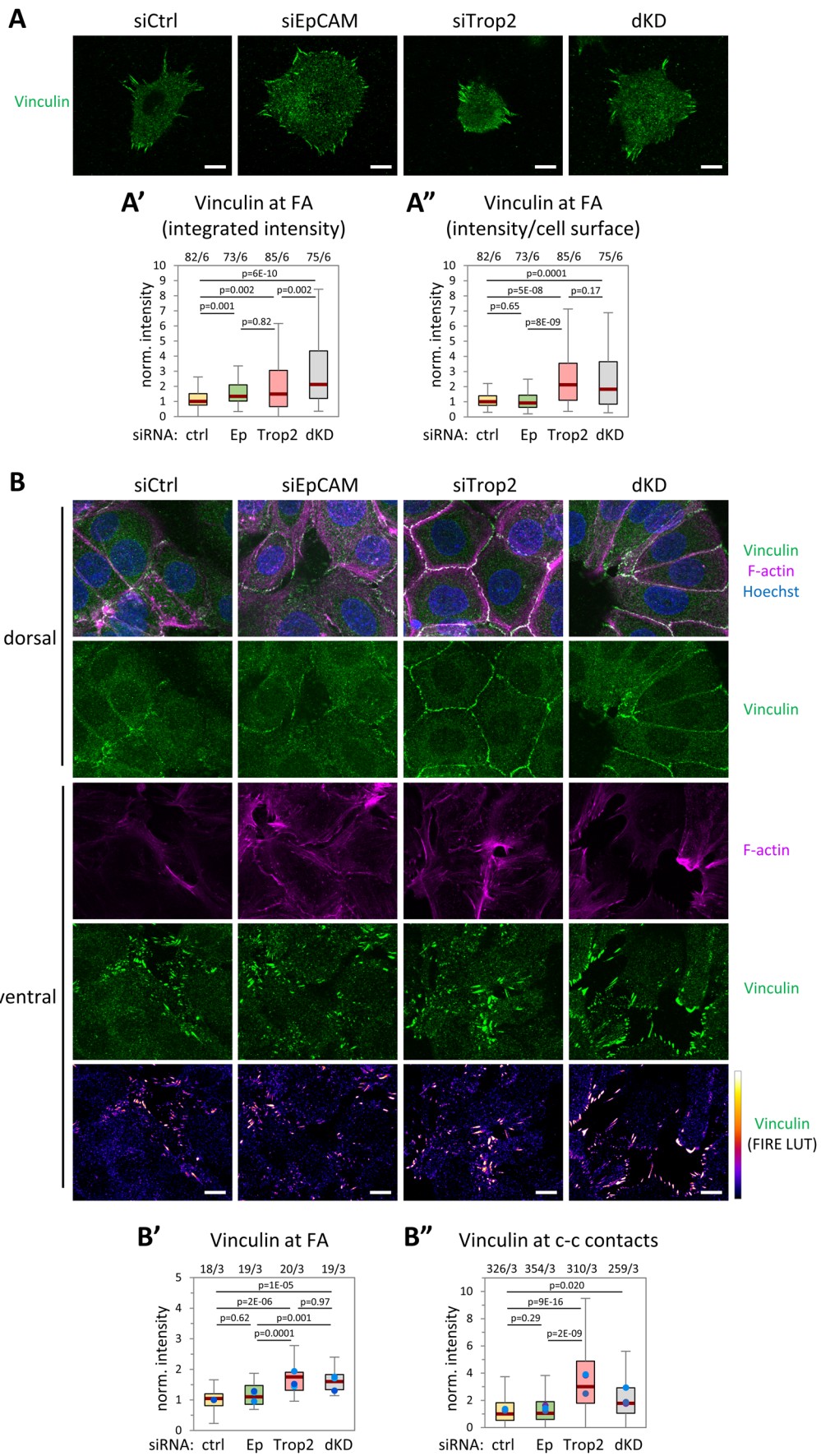

**Figure 3.  Different impact of EpCAM KD and Trop2 KD on recruitment of vinculin at focal adhesions and cell–cell contacts.**

(A) Confocal images of the ventral side (maximal projections) of single cells on collagen gel, immunolabelled for vinculin. (A') Quantification of integrated vinculin focal adhesion intensity, normalized relative to the median value of control cells. (A'') Vinculin signal normalized to the cell area. (B) Confocal images of large groups of cells laid on thin fibrillar collagen (~5-μm thick), immunolabelled for vinculin, and stained with phalloidin. Maximal projections of deconvoluted images from z-stacks, 1-μm thick for the ventral side, ~3 μm for the dorsal side. (B') Vinculin intensity at focal adhesions, normalized to the average of controls. (B'') Vinculin intensity along cell–cell contacts. The box plots show the interquartile range (box limits), median (center line), and min and max values without outliers (whiskers). Numbers above the graphs give number of cells (A', A''), fields (B'), or contacts (B'')/number of independent biological replicates. Dots indicate medians of each experiment. One-way non-parametric ANOVA (Kruskal–Wallis Test) followed by Dunn's post hoc test. Scale bars: 10 μm.

## EpCAM and Trop2 depletions both lead to the upregulation of myosin activity, but have distinct effects on the local recruitment of myosin, cadherin, and vinculin

Next, we characterized the effect of EpCAM and Trop2 depletions on the actomyosin cortex and adhesive structures. We conducted this analysis on whole spheroids, on small groups of cells, and on single cells, in order to relate behavior at the tissue scale to intrinsic cellular and biophysical properties.

Starting with spheroids, we assessed the impact on the cell cortex by probing for phosphorylated myosin light chain (pMLC) and F-actin (phalloidin), and on E-cadherin levels at cell–cell contacts. A first observation was that both EpCAM KD and Trop2 KD led to a mild but highly reproducible increase in global cortical pMLC (Fig. 2A,C). We conclude that KD of either EpCAM or Trop2 indeed led to higher cortical myosin activity, as expected from their role as myosin repressors. Another shared effect of both depletions was an enhanced E-cadherin signal (Fig. 2B,E), in keeping with the cell adhesions responding to increased tension by recruiting more cadherins (Engl et al, 2014; Gao et al, 2018). However, beyond these general effects common to both depletions, we observed striking differences in pMLC distribution: In EpCAM KD, pMLC was enriched by ~50% along the free edge of the spheroids, which was not the case for Trop2 KD (Fig. 2A,D; Appendix Fig. S5A). Phalloidin staining confirmed that EpCAM KD spheroids had a more robust peripheral cortex (Fig. 1G; Appendix Fig. S5A,B). The differences had important implications when put in the context of the abovementioned principle of balance of cortical tensions. In EpCAM KD spheroids, higher contractility along free edges should favor cell–cell adhesion and strengthen tissue cohesion (Fig. EV1), also explaining the smooth contours (Fig. 1E'). On the contrary, higher pMLC at cell contacts of Trop2 KD cells was predicted to result in lower cohesiveness, in agreement with the ragged outlines of these spheroids. That distinct adhesive states were achieved was further apparent from the pMLC/E-cadherin ratio, which was lower for EpCAM KD and higher for Trop2 KD (Fig. 2F).

We evaluated the same parameters on small groups of MCF7 cells (Fig. 2G–L). The groups ranged from 2–6 cells, which provided configurations where each cell in a group had at least one neighbor, was in contact with the matrix, and had a surface exposed to the medium (see orthogonal slice Fig. 2G'''). We measured peak intensity for each interface, which gave a robust readout over the large diversity of geometries observed for these cell groups. In controls, the pMLC signal was moderate at the free edges, and low, often undetectable at cell–cell contacts (Fig. 2G', concave yellow arrowheads), consistent with the well-established downregulation of tension along adhesive contacts (Maître and Heisenberg, 2013;

Winklbauer, 2015 and Fig. EV1). The effect of EpCAM and Trop2 KD were globally in line with those of spheroids. Firstly, both depletions increased cortical pMLC at edges and at cell–cell contacts, confirming the shared role of the two molecules in downregulating contractility (Fig. 2H,I). Yet, EpCAM KD had again a higher impact at edges relative to contacts, while the trend was opposite in Trop2 KD (Fig. 2H,I). The resulting ratio between contacts and edges was strongly increased in Trop2 KD, but not in EpCAM KD (Fig. 2J). Both depletions also enhanced E-cadherin at contacts (Fig. 2G,K). Here, Trop2 KD yielded the most intense signal but also again the highest pMLC/E-cadherin ratio (Fig. 2L). We also quantified the phalloidin signal, which yielded similar results (Appendix Fig. S6).

In summary, data from spheroids and small cell groups largely coincided. The fact that both depletions led to increased pMLC supported a common role of EpCAM and Trop2 in moderating cortical myosin, in agreement with the known function of Xenopus EpCAM (Maghzal et al, 2013). Furthermore, both depletions led to E-cadherin recruitment, indicating that adhesive contacts were, in both cases, capable to respond to increased cell contractility. Importantly, however, two potentially decisive differences emerged: Trop2 KD led to a stronger pMLC increase at contacts relative to edges, and to a higher pMLC to E-cadherin ratio, indicative of a fundamental difference in the balance of tensions, redirected more toward cell contacts in Trop2 KD.

We next analyzed cell–matrix contacts by quantifying vinculin. Vinculin is indeed recruited to focal adhesions (FAs) in a mechanosensitive manner, and its levels reflect tension exerted on FAs. For single cells, the total integrated FA signal per cell was increased in both EpCAM KD and Trop2 KD (Fig. 3A'), consistent with higher global tension due to hyperactivated actomyosin. However, when this signal was normalized relative to the ventral cell surface area (~vinculin "density"), it was significantly higher for Trop2 KD, while for EpCAM KD it remained similar to controls (Fig. 3A''). Thus, excess tension was distributed over an enlarged ventral interface in EpCAM KD cells, while Trop2 KD cells exerted a more acute tension on a reduced interface. FA vinculin intensity per area on large cell groups gave the exact same trend (Fig. 3B'). Note that because the deformed interface resulting from cell traction on thick gels prevented accurate FA segmentation, the analysis of cell groups was performed on thinner collagen (~5 μm). Cells sensed here the rigidity of the underlying glass, as apparent from the robust ventral stress fibers (Fig. 3B), which are much less prominent in cells and tissues adhering to a soft matrix, (Yeung et al, 2005; Doss et al, 2020, Figs. 1,2 and EV2B).

In the same cell groups, vinculin was also detected at cell–cell contacts, weak in controls and EpCAM KD, but significantly stronger in Trop2 KD (Fig. 3B,B''). As in FAs, vinculin is also

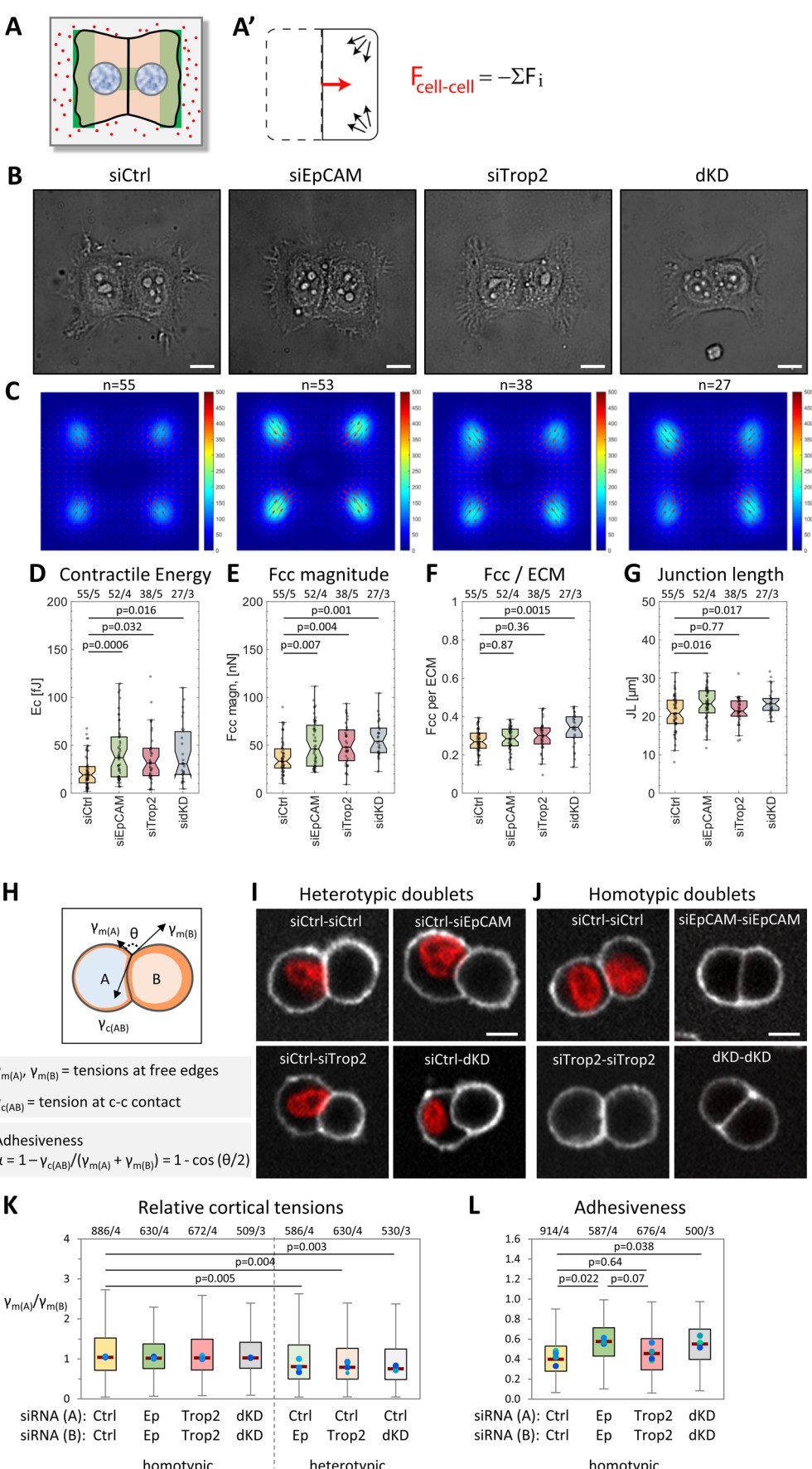

◀

**Figure 4. EpCAM KD and Trop2 KD differentially increase traction on substrate, cortical tension, and cell-cell adhesion.**

(A–D) Traction force microscopy (TFM) of cell doublets on H patterns. (A) Scheme of the experimental settings. Cells doublets were laid on H patterns coated with a thin layer of collagen (green) on a polyacrylamide gel with a stiffness of 5 kPa, containing far-red fluorescence nanobeads. (A') Diagram of force vectors obtained by TFM on H-shape micropatterns. Traction forces (black arrows) were measured from the displacement of nanobeads. The cell-cell force (red arrow) were calculated indirectly based on it counterbalancing the sum of traction forces, as shown in the equation. (B) Representative images of micropattern-confined cell doublets for the four experimental conditions. (C) Corresponding average maps of traction forces. (D–G) Quantification of (D) total contractile energy, Ec, (E) cell-cell forces (Fcc), (F) cell-cell force to traction force (ECM) ratio, and (G) junction lengths, measured from the phase contrast images. The plots show the interquartile range (box limits), median (center line), and min and max values without outliers (whiskers). Dots represent values for single doublets. Numbers of doublets and biological replicates above graphs. One-way ANOVA followed by Tukey-HSD post hoc test. Scale bars: 10 µm. (H–L) Determination of relative cortical tension and cell-cell adhesiveness of cell doublets laid on a non-adhering surface. (H) Diagram of an asymmetric cell doublet, with the balance between cortical tensions at the free edges ($\gamma_{m(A)}$ and $\gamma_{m(B)}$) and contact tension ($\gamma_{c(AB)}$) as introduced in Fig. EV1. The force vectors $\gamma_{m(A)}$, $\gamma_{m(A)}$, and $\gamma_{c(AB)}$ are tangential to the membranes at the cell vertex, which allows to directly calculate the relative strengths of these tensions based on the geometry at vertices. The orange layer represents the actomyosin cortex, with its thickness symbolizing relative contractility. A curved cell-cell interface reflects differences in cortical tension, as cell A tends to partly engulf B when $\gamma_{m(A)} < \gamma_{m(B)}$. θ, the angle formed by the two vectors $\gamma_{m(A)}$ and $\gamma_{m(B)}$, directly relates to adhesiveness, a dimensionless value that ranges from 0 to 1. (I, J) Examples of re-associated heterotypic doublets and homotypic doublets on an adhesion-free support, imaged by live confocal microscopy. Cell types indicated above each doublet. Membranes were labeled with CellMask Alexa Fluor 647. Hoechst 33342 was used to mark siCtrl cells in heterotypic doublets. Scale bar: 10 µm. (K) Quantification of relative cortical tensions expressed as the ratio $\gamma_{m(A)}/\gamma_{m(B)}$. As expected, the ratio is close to 1 for homotypic doublets under all conditions. The ratio for heterotypic doublets is significantly lower, demonstrating that single and double depletions all cause an increase in cortical tension. (L) Quantification of adhesiveness for homotypic doublets, calculated from $\gamma_c$ and $\gamma_m$. The box plots show the interquartile range (box limits), median (center line), and min and max values without outliers (whiskers). Experiment averages indicated by dots. Numbers of doublets and of biological replicates above graphs. One-way non-parametric ANOVA (Kruskal–Wallis Test) followed by Dunn post hoc test on experimental averages.

recruited in a tension-dependent manner by the cadherin-catenin complex (CCC), strengthening coupling with the actomyosin cortex (Leckband and de Rooij, 2014). Thus, cadherin contacts can respond to tension in two ways: Repressing local contractility and/or reinforcing linkage to the cortex (Maître and Heisenberg, 2013; Engl et al, 2014; Charras and Yap, 2018). The relative levels of vinculin relative to CCC can then provide information on the mechanical state of adhesive contacts. We performed double IF for β-catenin and vinculin, and quantified both signals at contacts (Fig. EV3). β-catenin levels increased in both EpCAM and Trop2 KD, confirming E-cadherin data (Fig. 2). However, the vinculin/β-catenin ratio was decreased in EpCAM KD and increased in Trop2 KD (Fig. EV3F). A closer look at these contacts showed that in controls and EpCAM KD, vinculin and β-catenin signals were largely complementary (Fig. EV3A'), vinculin typically high at sites of low β-catenin, but low along β-catenin-rich contacts. This pattern was blurred in Trop2 KD, with vinculin frequently accumulating along bright β-catenin membranes (Fig. EV3C', red arrows), further supporting that the adhesive sites were under higher burden.

In summary, a coherent body of data indicated that while depletion of either EpCAM or Trop2 boosted global myosin activation, the impact on cell interfaces was clearly different, indicative that both matrix and cell adhesions were under more tension in Trop2 KD.

Note that for consistency, we included dKD in most of these experiments. While some of the effects were bound to be complex and difficult to interpret, we highlight a few simple observations: dKD tended to give the highest intensities for total and edge pMLC (Fig. 2C,D,H), and for total vinculin (Fig. 3A), consistent with both EpCAM and Trop2 contributing to moderate cortical tension. Otherwise, dKD mostly gave intermediate values between those of the single KDs.

## Impact of EpCAM and Trop2 KD on cell-substrate and cell-cell forces

These observations prompted us to measure the actual forces exerted on different structures. We used traction force microscopy

(TFM) on cell pairs adhering to H-shaped micropatterns (Fig. 4A). The H patterns constrained the cell pair to adopt a stable configuration, providing a robust system to directly measure traction exerted on the matrix and, at the same time, to indirectly calculate the force at the cell-cell contact based on the balance of forces (Fig. 4A') (Tseng et al, 2012). In these settings, traction exerted on the matrix appears to be similar for single cells and for cell doublets (Ruppel et al, 2023). Another advantage was that the H pattern, by imposing a fixed geometry and a limited surface for the contacts to the matrix, reduced the wide differences in cell morphology of cells on non-constrained collagen substrate, naturally observed within the wild type MCF7 cell population, and further exacerbated by EpCAM and Trop2 depletions.

Using this approach, we found that traction on the matrix was stronger in both EpCAM KD and Trop2 KD, corresponding to higher total tensile energy (Fig. 4C,D). These results directly demonstrated the increased contractility implied by the pMLC IF data, and supported by increased total FA vinculin signal (Fig. 3A'). As for vinculin "density", higher for Trop2 KD than EpCAM KD (Figs. 3A",B' and EV3E), it is related to the difference in spreading, or "wetting" the substrate, which, does not depend on absolute contractility, but on the balance of tension between interfaces (Fig. EV1). Similarly, the calculated force exerted on the cell-cell contact (Fcc) was also increased in both depletions (Fig. 4E), but the length of cell-cell contacts, which is a readout of adhesiveness (Fig. EV1), was significantly broader for EpCAM KD pairs, while only marginally increased in Trop2 KD (Fig. 4B,G). As expected, dKD also increased contractile energy, yielding the highest cell-cell force and highest cell-cell to total force ratio, consistent with EpCAM and Trop2 both contributing to moderating myosin contractility in these cells (Fig. 4E,F).

One should here clarify that while these TFM experiments yielded unambiguous measurement on global tensile properties, they did not provide direct information on the balance of tensions. In particular, traction forces relate to - $\gamma_s$, thus both $\gamma_m$ and $\gamma_x$ (Fig. EV1A). Furthermore, H patterns force cells to adopt a highly constrained configuration, and organize actomyosin cables quite differently than for classical free migrating cells (Ruppel et al, 2023). Here, in particular, all cells were strongly spread, even siTrop2 KD cells, which otherwise are round and tall.

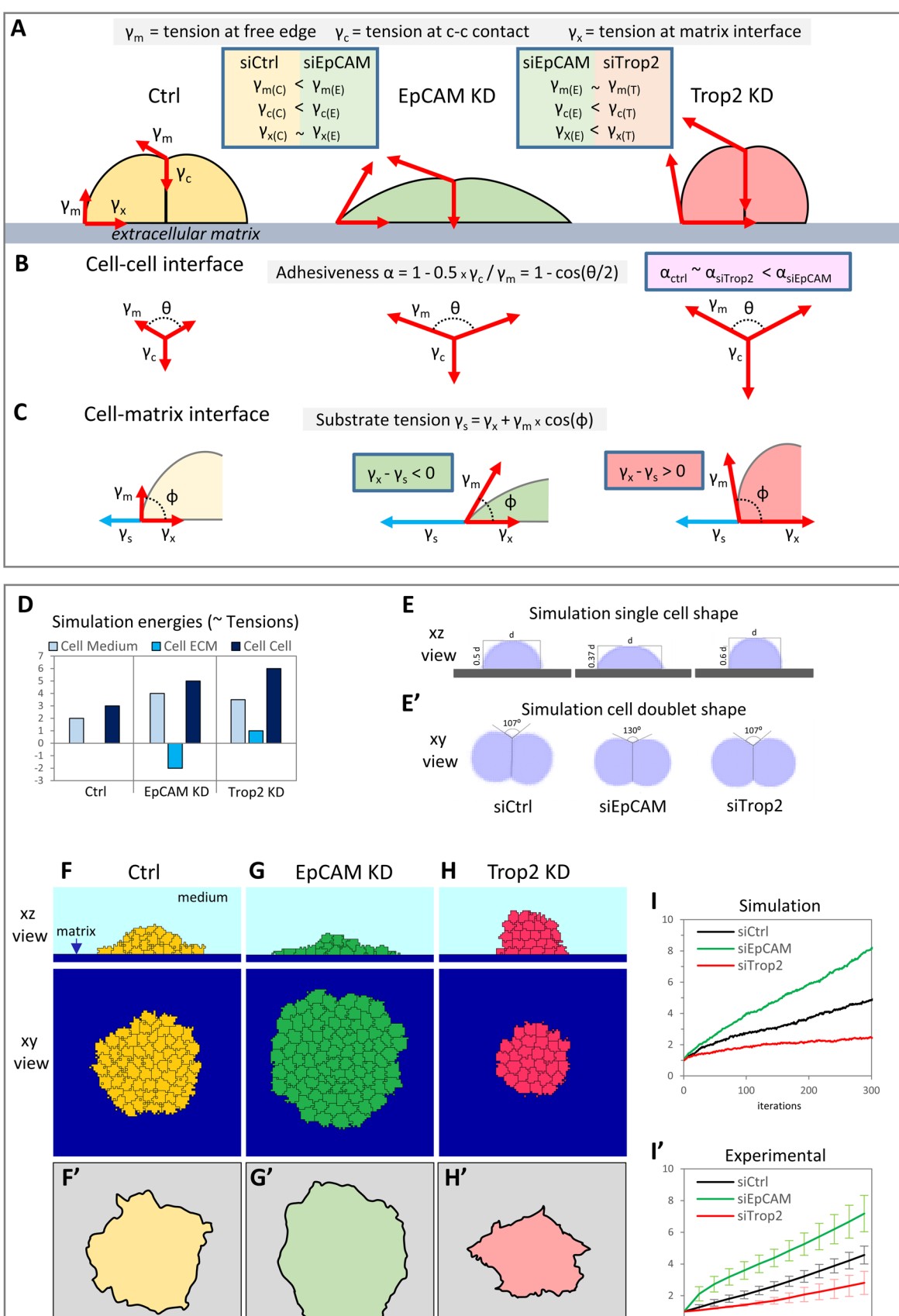

**Figure 5.  Simulation of MCF7 spheroid spreading in controls and under EpCAM/Trop2-depleted conditions based on biophysical parameters.**

(**A**) Schematic cell doublets on a substrate, with the relative tension vectors represented at the cell-cell vertex and at the edge of the matrix substrate interface (see Fig. EV1 for details). Their relative strength is represented by the length of the vector. The general changes in tensions between control and EpCAM KD, and between EpCAM KD and Trop2 KD are summarized in the two boxes with blue borders. (**B**) Balance of forces at the cell-cell vertex. While all three tensions are strongly increased both in EpCAM KD and Trop2 KD, the resulting adhesiveness is increased in EpCAM KD but not in Trop2 KD. (**C**) Balance of forces at the matrix interface. $\gamma_s$ is the tension exerted on the substrate. It counterbalances the cell-matrix tension $\gamma_x$ and the horizontal component of the cell-medium tension $\gamma_m$. Both tensions are increased in EpCAM and Trop2 KD, but their relative balance determines different outcomes: EpCAM KD cells adopt a spread configuration (acute angle Φ), while, on the contrary, Trop2 KD cells have a compact shape (obtuse angle Φ). (**D**) Energy parameters were used to simulate the experimental conditions (see Appendix Section 1). (**E, E'**) Simulation of single cell and cell doublet geometry obtained using a 3D cellular Potts model (CompuCell3D). (**E**) Profile view of a single cell. The numbers correspond to the height/ diameter ratio (assuming a circular cell surface) from experimental data (Fig. 1I). (**E'**) Top view of average non-adhesive cell doublets based on experimental data (Fig. 4H, L). (**F–H**) Simulation of spheroids spreading for control, EpCAM KD and Trop2 KD conditions. Images correspond to examples of xy and yz planes after 300 iterations. Medium is in light blue, and matrix substrate is in dark blue. (**F'–H'**) Outlines of explants at 24 h from Fig. 1, drawn for comparison. (**I, I'**) Comparison of simulated relative area expansion with average values from experiments presented in Fig. 1. Error bars: SD.

The substrate used for TFM was an acrylamide gel coated with collagen. We verified the effect of EpCAM KD on spreading of spheroids on this acrylamide-based substrate. We found essentially the same phenotypes as on fibrillar collagen gel: EpCAM depletion strongly increased spreading, while maintaining high cohesiveness as indicated by high solidity values (Appendix Fig. S7).

## Impact of EpCAM and Trop2 KD on cortical tension and cell-cell adhesiveness

An unequivocal method to determine differences in cortical tension and in cell-cell adhesiveness is to perform force inference based on the contact vertex geometry of free cell doublets on a non-adherent surface (David et al, 2014; Winklbauer, 2015; Canty et al, 2017; Parent et al, 2017; Fagotto, 2020b). Such doublets adopt a typical configuration, where the cell contact expands until contact tension $\gamma_c$ is precisely balanced by the cortical tension at the interface with the medium $\gamma_m$. As a consequence, a larger contact angle θ directly relates to higher adhesiveness (Figs. 4H and EV1). Moreover, if cell A and cell B have different cortical tensions $\gamma_{m(A)}$ and $\gamma_{m(B)}$, the heterotypic doublet is asymmetric, and the softer cell tends to engulf the stiffer cell (Fig. 4H). Thus, the geometry of the contact vertex directly reflects the balance between the three tensions, $\gamma_{m(A)}$, $\gamma_{m(B)}$, and $\gamma_{c(AB)}$ (Canty et al, 2017; David et al, 2014; Kashkooli et al, 2021; Fagotto, 2020b).

We adapted the protocol previously used for embryonic cells (Rohani et al, 2014; Kashkooli et al, 2021) to MCF7 monolayers for gentle dissociation into single cells. We then mixed at low-density control and depleted populations, imaged both homotypic and heterotypic doublets, and determined the relative tensions (Fig. 4I,J). These experiments led to the following conclusions: Firstly, heterotypic doublets made of a control (A) and a depleted cell (B) were clearly asymmetric (Fig. 4I), resulting in $\gamma_{m(A)}/\gamma_{m(B)}$ significantly lower than 1 (Fig. 4K). This constituted a firm demonstration that cortical contractility was indeed increased upon EpCAM and Trop2 KD. Second, contact angles θ of homotypic doublets were less acute for EpCAM KD than controls (Fig. 4J), corresponding to a significant increase in adhesiveness (median 0.58 compared to 0.40 for controls (Fig. 4L). In the case of Trop2 KD, however, only a small, non-significant increase was observed (Fig. 4L). These two results were in perfect agreement with the results of TFM, with spheroid geometry, and with pMLC, E-cadherin and vinculin IF. Taken together, these data showed that cell-matrix and cell-cell interfaces were impacted in similar

ways: Both EpCAM KD and Trop2 KD led to higher forces exerted on the two types of contacts, but adhesiveness and wetting were only increased in EpCAM KD, not in Trop2 KD.

## Spheroid spreading phenotypes can be simulated based on cell tensile properties

With this comprehensive characterization of the EpCAM KD and Trop2 KD phenotypes in hand, we asked whether the properties observed at the cell level could account for the spheroid configurations. We modeled the system based on the balance of the tensions exerted along the three cell interfaces $\gamma_m$, $\gamma_c$, and $\gamma_x$ (EV1). It is important to keep in mind that the extent of spreading of a cell on the substrate ("wetting") does not depend on absolute tension strengths, but on their ratios. The three configurations shown in Fig. 5A–C roughly correspond to wildtype, EpCAM KD and Trop2 conditions, and illustrate how cells and tissues undergoing a rise in global contractility are predicted to adopt radically opposite configurations depending on the distribution of tensions. Indeed, EpCAM KD resulted in the expansion of contact interfaces, which compensated for increased tension and led to higher adhesiveness and spreading. Trop2 KD, on the contrary, produced configurations with smaller contact areas, which proportionally were subjected to more acute tension (Fig. 5A–C).

Using the framework, we simulated spheroid spreading using a 3D version of a cellular Potts model (CompuCell3D (Izaguirre et al, 2004; Cickovski et al, 2005). In this model, the tensions for a given cell type are input as relative "energies" (see Appendix Section 1). These energies are related to $\gamma_m$ for the cell-medium energy, $\gamma_c$ for cell-cell energy, and $\gamma_x$-$\gamma_s$ for cell-matrix energy. The key features were (1) higher $\gamma_m$ for both EpCAM KD and Trop2 KD, which accounted for increased contractility compared to controls, (2) a lower cell-cell energy/cell-medium energy ratio (thus higher adhesiveness) for EpCAM KD, as well as (3) a cell-substrate energy $\gamma_x$ set negatively for EpCAM KD ($\gamma_x$-$\gamma_s$ < 0) and positively for Trop2 KD ($\gamma_x$-$\gamma_s$ > 0). Note that these "tensions" are single terms that integrate various positive and negative inputs. For instance, $\gamma_x$ is not only determined by the properties of the cell cortex, but also by the contribution of the ventral stress fibers.

We set the three energies (Fig. 5D) to fit the geometry of single cells and cell doublets (Figs. 1K, 4H,L and 5E, see Appendix Section 1), which directly relates to the balance of tensions. While TFM data could not be input as such in the model, we did take into account TFM measurements of total contractile energy to tune

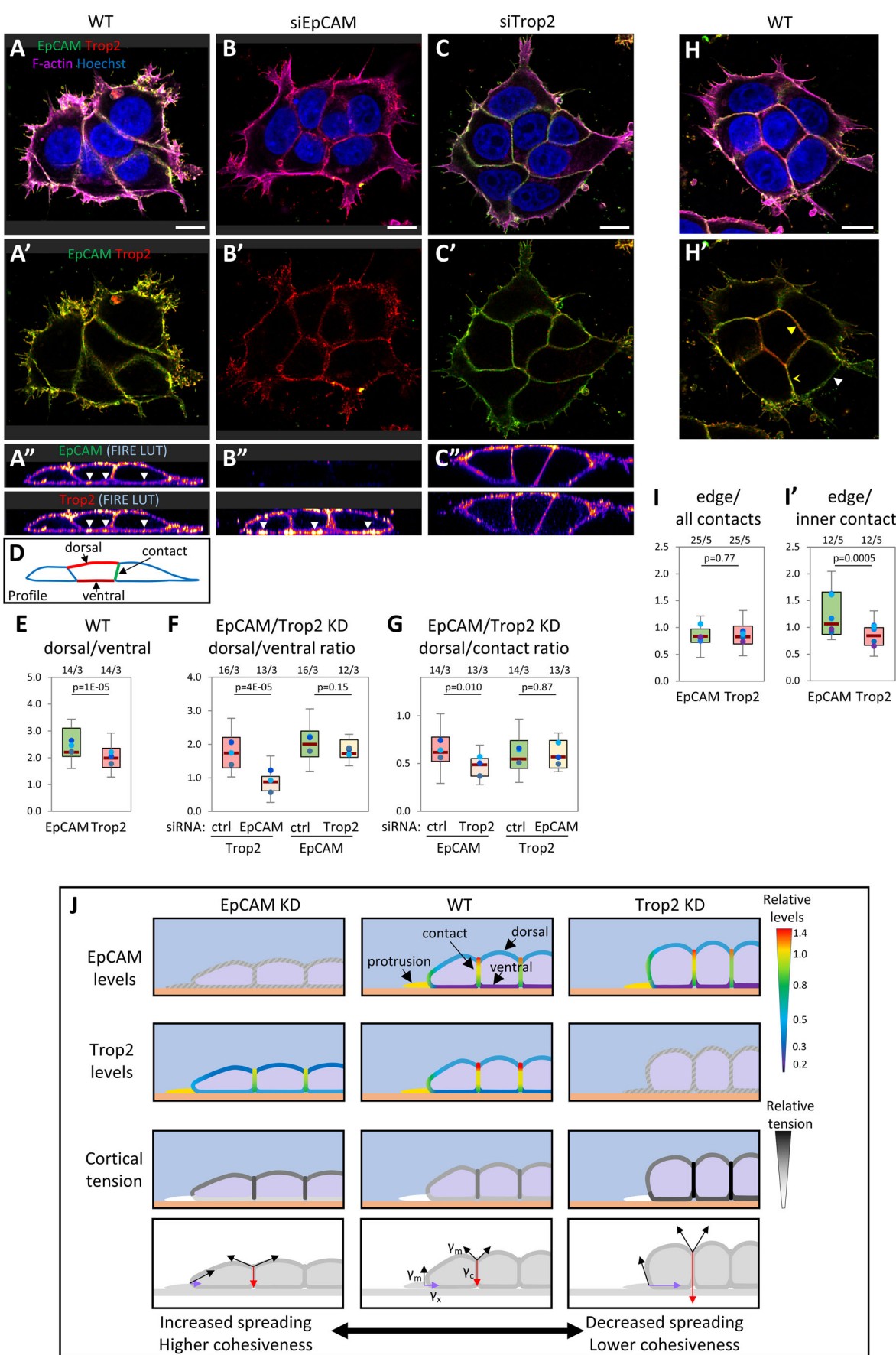

**Figure 6. EpCAM and Trop2 differential distribution along cell interfaces of MCF7 cells and impact of depletions.**

(A–C) Confocal microscopy image of small groups of wild type, EpCAM KD and Trop2 KD cells on collagen gel, labeled under non-permeabilized conditions for EpCAM (green), Trop2 (red) and F-actin (phalloidin-Alexa647, magenta). (A–C) Horizontal plane, merged image, maximal projection of 4 slices, 1 µm total. (A'–C'). Merge of only green (EpCAM) and red (Trop2) channels. (A"–C") Profile, orthogonal views of the same groups. EpCAM and Trop2 channels are displayed as "FIRE" LUT. Arrowheads in (A") and (B") point to the slight enrichment of Trop2 compared to EpCAM in the ventral side. (D–F) Quantification. EpCAM and Trop2 levels where analyzed from whole z-stacks, as well from resliced stacks to obtain profile projections (details in Appendix Fig. S8). (D) Diagram of the three main interfaces quantified based on profiles. (E) Pairwise comparison of dorsal and ventral interfaces for WT groups. (F, G) Dorsal/ventral and dorsal/contact ratios for controls and EpCAM or Trop2 KD. See Appendix Fig. S8 for more detailed quantifications. The box plots show the interquartile range (box limits), median (center line), and min and max values without outliers (whiskers). Number of cell groups/biological replicates is indicated above graphs. Statistical analysis: Pairwise two-tailed Student's t-test. (H) Example of cell group with a central cell, surrounded by "inner" contacts (yellow arrowhead). Yellow concave arrowheads point to an outer contact, and white arrowheads to the free edge. (I, I') Ratios edges/all contacts (I) and edges/inner contacts (I'), for EpCAM and Trop2. Quantification of 28 groups of 6–12 cells, from five experiments, of which 13 groups had inner contacts. Statistical analysis: Pairwise two-tailed Student's t-test. Scale bars: (A–C), 5 µm; (H) 10 µm. (J) Summary diagram of EpCAM and Trop2 distribution in wild type and depleted MCF7, and impact on cortical tensions. The schemes represent three cells at the edge of a group. The two top rows of panels use a color code to recapitulate the relative levels of EpCAM and of Trop2 under wild type and EpCAM/Trop2-depleted conditions. The third row provides a summary of cortical tension at the various interfaces using an arbitrary grayscale. The lower row represents the three corresponding cellular tensions $\gamma_m$ (black), $\gamma_c$ (red), and $\gamma_x$ (purple).

relative cell-medium energy values, which seemed a reasonable assumption. We should emphasize that the selected energy values, although approximative, were coherent with the ensemble of our experimental data.

Simulation of the behavior of spheroids under the three conditions yielded clear differences in the degree of spreading, moderate for the control condition, high for EpCAM KD, and low for Trop2 KD (Figs. 5F–I; Movies EV5 and 6). Thus, inputting simple parameters inferred from experimental data on single cells/ doublets faithfully reproduced the experimental spheroid spreading phenotypes (Fig. 5F'–I'). Spheroids continued to spread past the 24 h time point used in most of our experiments. We thought to confront our Potts simulation with the morphology of spheroids after 48 h (Fig. EV4A). Notably, EpCAM KD spheroids achieved then maximal wetting of the substrate, forming a flat monolayer that remained fully coherent, as judged by tight cell-cell contacts, a smooth dorsal surface, and a smooth edge (Fig. EV4B,B'). When run for 600 iterations (twice longer than for Fig. 5), the model reproduced remarkably well these late configurations (Fig. EV4D). The only major feature that was poorly reproduced was the convoluted edges typical of Trop2 KD, both at 24 and 48 h. We concluded from these stimulations that the observed differences in contractility at different cell interfaces were sufficient to account for the diametrically opposite spreading behaviors produced at the tissue level by EpCAM and Trop2 depletions.

## Partial differential distribution of EpCAM and Trop2 at cell interfaces may account for the distinct phenotypes

The simplest possibility to account for the opposite phenotypes of EpCAM and Trop2 KD was a difference in distribution of the two regulators, which may then preferentially repress myosin on one or the other of the cortical interfaces. We set up to determine these distributions by quantitative IF on small groups of cells. Both EpCAM and Trop2 were strongly expressed all along the plasma membrane, accounting for their global impact on contractility and adhesiveness. Standard IF with post-fixation permeabilization detected few EpCAM and/or Trop2-positive intracellular spots, that represented a negligible pool relative to the surface signal (Appendix Fig. S8E). Because permeabilization led to a significant loss of Trop2 signal at the plasma membrane, we present data from cell surface labeling without permeabilization, which gave robust signals for both proteins (Fig. 6A). We quantified their distribution

along the ventral side in cell-matrix contacts, the cell-cell contacts, as well as the dorsal side/lateral free edge, excluding protrusions that were measured separately (diagrams Fig. 6D; Appendix Fig. S8B). These locations corresponded to the sites where the three basic cortical tensions $\gamma_x$, $\gamma_c$, and $\gamma_m$ are exerted. While the two molecules showed a very similar general distribution (Fig. 6A; Appendix Fig. S8B), we found a conspicuous quantitative difference between the free surface (dorsal side and lateral edges) and the ventral interface, the latter displaying proportionally more Trop2 than EpCAM (Fig. 6A",E). The resulting difference in dorsal/ ventral ratio was modest (~20%) but reproducible (Fig. 6E). A more detailed analysis that included additional subregions confirmed this (Appendix Fig. S8A–C). A second difference was observed specifically for cell-cell contacts at the interior of cell groups (here named inner contacts), where Trop2 was also enriched relative to EpCAM (Fig. 6H,I, example of larger group in Appendix Fig. S8D). Loss-of-function situations led to a more contrasted landscape: Although EpCAM levels and distribution were largely unaffected in Trop2 KD cells, Trop2 was significantly impacted by EpCAM KD; its levels decreased on the whole lateral and dorsal free edges, while they increased at the ventral interface (Fig. 6B,C; Appendix Fig. S8C), resulting in a strong reduction in both the free edge/ ventral ratio, and the free edge/cell-cell contact ratio (Fig. 6F,G). These changes were of importance for interpreting the EpCAM KD phenotypes, since they were bound to exacerbate the tension imbalance caused by EpCAM loss, further increasing edge tension $\gamma_m$, while mitigating the impact on contact tensions $\gamma_x$ and $\gamma_c$. This situation was perfectly consistent with EpCAM KD boosting both spreading on the matrix and tissue cohesion. The case was the opposite for Trop2 depletion, which was predicted to deregulate more acute tension along the ventral interface and cell-cell contacts, explaining reduced tissue spreading and cohesion. The distributions of EpCAM and Trop2 at various cell interfaces and the predicted impact of their depletions on the balance of cortical tensions and on tissue cohesion and spreading are schematically summarized in Fig. 6J.

## Differential LOF phenotypes and differential distribution of EpCAM and Trop2 in other types of breast cancer cells

We asked whether similar differences in EpCAM and Trop2 could also be detected in other cell lines. We selected two additional luminal breast cancer cell lines, HCC1500 and MDA-MB-453, that

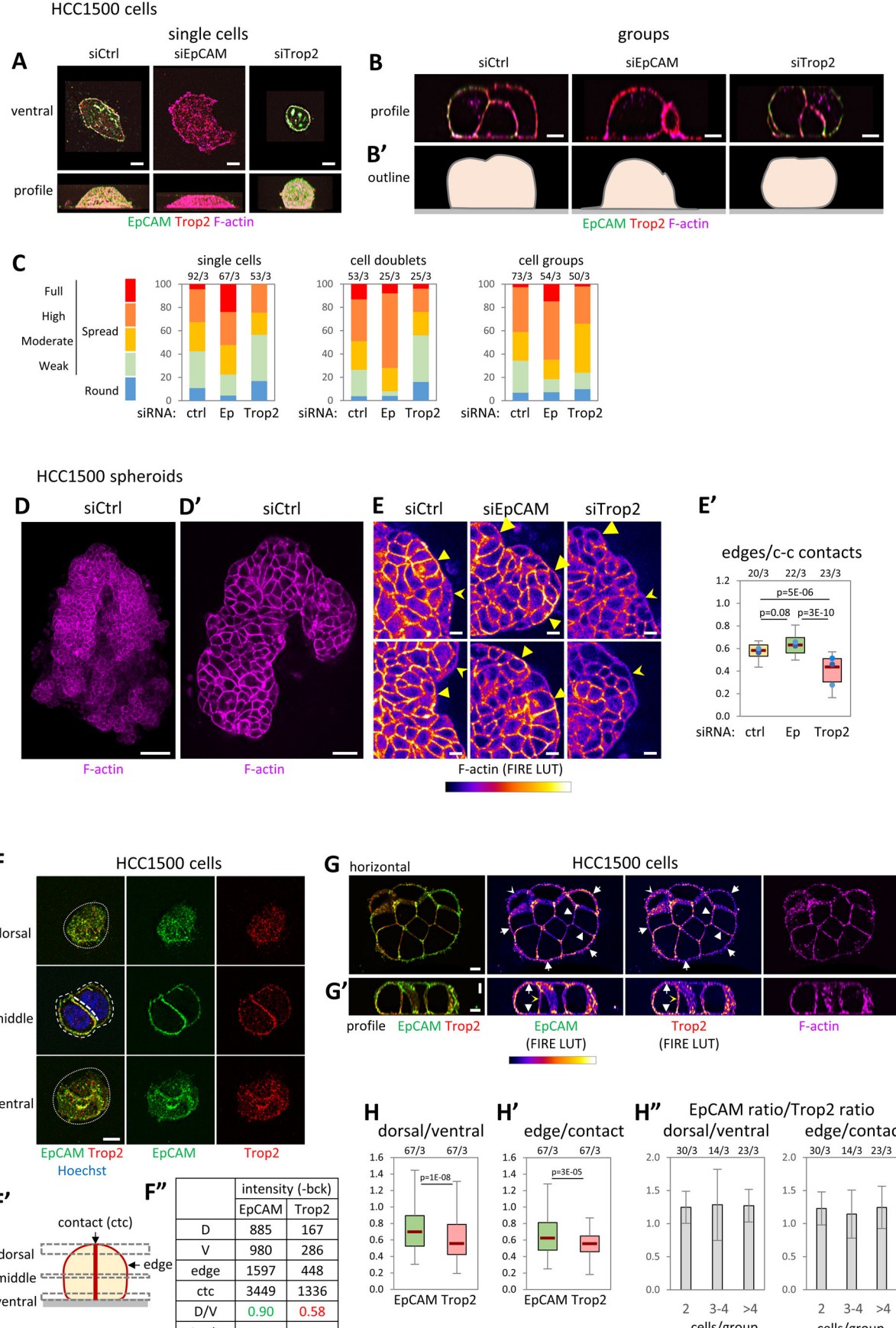

◄

**Figure 7. EpCAM KD and Trop2 KD phenotypes in other breast cancer cell lines.**

(**A–C**) EpCAM KD and Trop2 KD in HCC1500 cells differentially impact substrate wetting. HCC1500 cells seeded on a collagen gel were cell surface immunolabelled for EpCAM and Trop2, stained with phalloidin, and imaged by confocal microscopy. The morphological analysis was made on profiles from orthogonal views, indicative of the degree of wetting, as detailed in Figure EV5. (**A, B**) Examples of single cells in (**A**) and groups in (**B**). See Fig. EV5 for multiple views of the same cells and cell groups. **B'** highlight the outlines of the groups. Note that the extreme spreading observed for some single EpCAM KD cells shown in (**A**) was rarely found in cell groups. (**C**) Quantification. The bar graphs show compiled results for single cells, cell doublets, and groups of cells, which all show similar trends. Numbers of cells or cell groups/ biological replicates indicated above graphs. See Fig. EV5D for statistical analysis. Scale bars, 5 µm. (**D, E**) EpCAM KD and Trop2 KD differentially impact the cortex at free edges and cell-cell contact in HCC1500 spheroids. Spheroids formed from control, siEpCAM and siTrop2 cells were stained with phalloidin and imaged by confocal microscopy. (**D**) Overview of a control spheroid, presented as maximal projection. HCC1500 spheroids are typically highly convoluted. (**D'**) Optical slice of the same spheroid. (**E**) Detailed views of spheroid edges, two examples per condition. Arrowheads point to edges to be compared with cell-cell contacts: Large filled, small filled, and concave arrowheads indicate respectively high, moderate, and low F-actin. (**E'**) Quantification of the edge/contact intensity ratio. Dots indicate averages for each experiment. One-way non-parametric ANOVA (Kruskal–Wallis test) followed by Dunn post hoc test. Scale bars: (**D**), 100 µm; (**D'**), 50 µm; (**E**), 10 µm. (**F–H**) Differential EpCAM and Trop2 distribution in HCC1500 cells. HCC1500 cells were surface immunolabelled for EpCAM and Trop2, and stained for Hoechst (**F**) or phalloidin (**G**). (**F**) Cell doublet. Three horizontal slices correspond to dorsal (0.2 µm max projection, 2 planes), middle (0.8 µm max projection, 5 planes), and ventral (0.2 µm max projection, 2 planes) regions (**F'**). Dotted lines indicate the areas used for quantification of dorsal and ventral signals. The dashed lines mark the free edge and cell contact interfaces, which were quantified through line scans on z profiles. Scale bar: 5 µm. (**F"**) Quantification for this doublet, providing average intensities, background subtracted, for dorsal and ventral surfaces, and for the lateral edge and cell-cell contact (ctc), and the resulting dorsal/ventral (D/V) and ctc/edge ratios. (**G**) Group of cells labeled for EpCAM, Trop2 and F-actin (phalloidin). Arrows point to edges with higher EpCAM relative to Trop2, arrowheads to contacts with low EpCAM and higher Trop2. Small concave arrowhead: Case of edge with high Trop2. (**G'**) Orthogonal view, used for quantification. White arrow points to dorsal edge, with high EpCAM, white arrowhead to the ventral interface, and yellow concave arrowhead to cell contact, both with higher Trop2. Horizontal and vertical scale bars: 5 µm. (**H, H'**) Quantification of dorsal/ventral and edge/contact ratios for EpCAM and Trop2 intensities, for groups ranging from doublets up to 20 cell-large groups. The box plots show the interquartile range (box limits), median (center line), and min and max values without outliers (whiskers). Numbers of cells/biological replicates indicated above graphs. (**H"**) Calculated EpCAM ratios divided by Trop2 ratios for groups of different sizes, showing that the relative enrichments are constant over the whole range. Error bars: SD. Statistical analysis using two-tailed Student's *t*-test.

express both EpCAM and Trop2 (Appendix Fig. S1), but differ in their morphologies and have behaviours departing in opposite directions from MCF7 cells: When grown on collagen, HCC1500 cells spread very poorly on matrix (Figs. 7A,C and EV5), but establish extensive cell-cell contacts and form coherent aggregates (Figs. 7B and EV5). The highly stereotypical morphology of single cells, groups, and spheroids (poor spreading, tall compact morphology, Figs. 7A,B,E and EV5A,B) was clearly indicative of high tension at both matrix and cell contacts relative to free edges. On the contrary, MDA-MB-453 rapidly spread and migrate, and, while capable of cell-cell adhesion, form less extensive cell contacts (Fig. 8A,B). Such configuration was predicted to correspond to a lower ratio between tension at contact interfaces and at the free edge.

EpCAM KD and Trop2 KD showed opposite phenotypes, in both cell types. Spreading of HCC1500 cells, single or in groups, was stimulated by EpCAM KD, but diminished upon Trop2 KD (Figs. 7A–C and EV5). HCC1500 spheroids did not spread under any condition (Fig. 7D). However, we observed differences at cortical interfaces, monitored by phalloidin staining, with the intensity ratio between free edges and cell-cell contacts highest for EpCAM KD and lowest for Trop2 KD (Fig. 7E). Contrary to HCC1500, wild type MDA-MB-453 spheroids adhered and spread rapidly and more extensively than MCF7 spheroids (Fig. 8A). After 18 h, a large portion of the cell mass had already migrated away from the original core (Fig. 8A,B), typically forming a loose monolayer (Fig. 8B'), the remnant of which still emerged as a dome (Fig. 8A–C). EpCAM or Trop2 KD did not significantly change global spheroid expansion (Fig. 8C"), but clearly impacted on spheroid morphology: Upon EpCAM KD, the spheroids spread into a flatter disc, with the central dome less prominent or absent (Fig. 8A,B,B',C'). Furthermore, cells near the edge maintained tight contacts, suggesting that the spheroid expanded as a more coherent mass (Fig. 8B'). In Trop2 KD, the core of the spheroid remained more massive, surrounded by a poorly coherent monolayer of cells (Fig. 8A,B,B',C'), suggesting that spheroid "spreading" was mostly

due to migration of loosely connected individual cells, while the core failed to expand. Altogether, despite the large differences in basal characteristics of HCC1500 and MDA-MB-453, the effects observed for EpCAM KD and Trop2 KD were all consistent with a differential action at different interfaces, similar to MCF7 cells.

We determined the cell surface distribution of EpCAM and Trop2 in wild-type HCC1500 and MDA-MB-453 cells. In HCC1500 cells, both EpCAM and Trop2 were globally enriched on the ventral side (dorsal to ventral ratio <1, contrary to >1 for MCF7 cells). Nevertheless, just like MCF7 cells, this D/V ratio was also significantly higher for EpCAM than for Trop2, and the same trend was found when comparing lateral edges to cell-cell contacts (Fig. 7F–H). In MDA-MB-453 (Fig. 8D,E), the differential distribution was most striking, EpCAM being clearly enriched on the dorsal side (D/V ratio >1), Trop2 on the ventral side (D/V ratio <1). Thus, similar biases between EpCAM and Trop2 could also account for EpCAM KD favoring wetting and Trop2 having the opposite effect in these breast cancer lines.

## Discussion

We uncovered in this study a subtle regulatory system that has a major impact on the adhesive and migratory properties of cell aggregates. EpCAM and Trop2 are very closely related proteins, and we validated here that when expressed in the same cells, they both acted as negative regulators of myosin activity to moderate global cell contractility. This was evident from the quantification of cortical pMLC and firmly demonstrated by our biophysical measurements. Nevertheless, EpCAM and Trop2 appeared to have diametrically opposite impacts at the cell and tissue level, EpCAM acting as a repressor of adhesion and collective migration, and conversely, Trop2 as an activator of these same processes, indicating that the two molecules function as a mechanostat in tissue morphogenesis. While we have discovered and characterized in detail this surprising phenomenon in MCF7 cells, we also

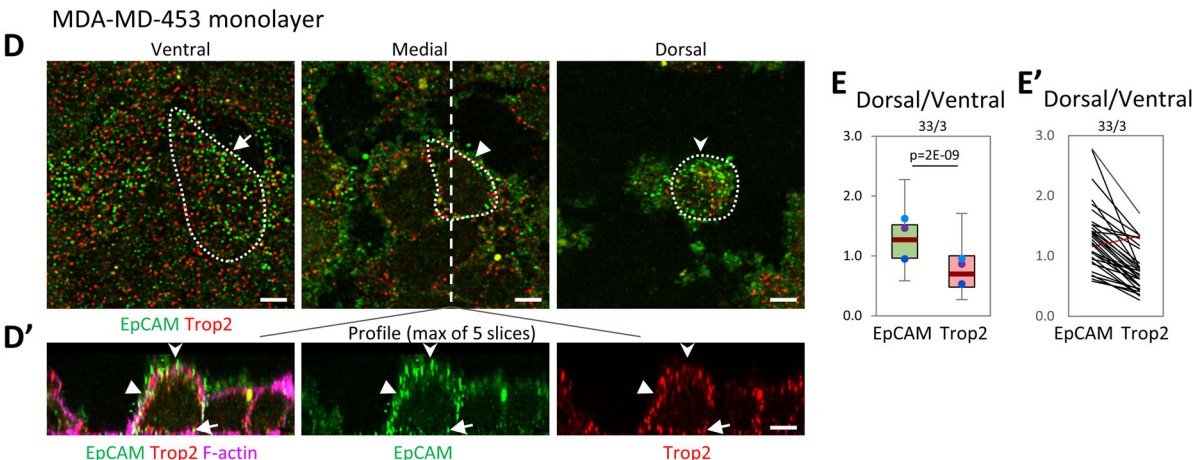

Figure 8.  EpCAM and Trop2 KD opposite phenotypes on MDA-MD-453 spheroids.

(A, B) Spreading and morphology of MDA-MD-453 spheroids. (A) Bright field images of control, EpCAM KD and Trop2 KD spheroids, shortly after adhering to the collagen gel ($t = 30$ min), and after 18 h. All spheroids have extensively spread. In control spheroids, a remnant core of the original cell mass can still be seen as a dome (arrow). The dome is much smaller (arrowhead) or absent in EpCAM KD spheroids, but typically larger in Trop2 KD spheroids. (B) Maximal projection of phalloidin-labeled spheroids. (B') From top to bottom: Profiles obtained from yz and xz orthogonal projections, scheme of xz profile, with red dashed line indicating the emerging dome, and detail of edge. White dashed line, topography of dorsal surface; arrowheads, very thin, loose contacts. Blue dashed boxes in the schemes correspond to the position of the enlarged image (xy detail). The red dashed lines mark the emerging core of the spheroid. (C) Schematic top and profile views of spread MDA-MB-453 spheroids. (C') Quantification of the relative core area (ratio to total spheroid area) after 18 h of spreading. The box plots show the interquartile range (box limits), median (center line), and min and max values without outliers (whiskers). Numbers of spheroids/biological replicates are indicated above graphs. (C") Area increase, not significantly different between conditions. One-way non-parametric ANOVA (Kruskal–Wallis test) followed by Dunn post hoc test. (D, E) EpCAM and Trop2 localization in wild-type MDA-MD-453 cells, plated on a collagen gel. EpCAM and Trop2 signals were variable and generally low. (D) Confocal slices of ventral, medial, and dorsal planes, with outlines of one cell indicated by a dotted line. (D') Profile obtained from orthogonal slices, maximum projection of five slices. Arrow: ventral interface; filled arrowhead: lateral surface; concave arrowhead: dorsal side. (E) Dorsal/ventral ratio of EpCAM and Trop2 intensities shows that EpCAM is enriched dorsally and Trop2 ventrally. Numbers of cells/biological replicates indicated above graphs. Dots: Averages of individual experiments. (E') Individual pairs of ratios for each cell. The red line indicates the only cell where the ratios were inverted. Statistical analysis using two-tailed Student's t-test. Scale bars: (A, B), 50 μm; (D), 5 μm.

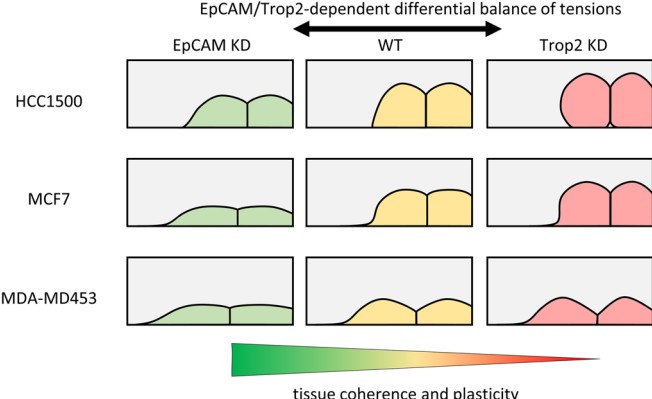

EpCAM/Trop2-dependent differential balance of tensions

EpCAM KD | WT | Trop2 KD

HCC1500

MCF7

MDA-MD453

tissue coherence and plasticity

**Figure 9.  Summary of EpCAM and Trop2 KD phenotypes in the three breast cancer cell lines tested in this study.**

HCC1500, MCF7, and MDA-MD-453 are all luminal breast cancer cell lines that share in common the ability to form E-cadherin cell-cell contacts, but differ in their ability to spread on a collagen matrix ("wetting"). Cell-cell adhesion, thus tissue cohesion, appears to dominate over matrix adhesion for HCC1500, while for MDA-MD-453 the balance is tilted the other way. MCF7 represents an intermediate case. In all three cell lines, EpCAM KD tends to stimulate both spreading on the matrix AND cell-cell adhesion/tissue cohesion (flatter angles at contact vertices), while Trop2 KD tends, on the contrary, to increase tension at both types of interfaces.

presented confirming evidence in two other breast cancer cell lines, which suggests that this is a general property of this pair of molecules.

The most striking phenotype resulting from EpCAM or Trop2 KD is increased, respectively decreased spreading, consistently observed for single cells, cell groups, and spheroids, and in all three cell lines. This effect is clearly linking EpCAM and Trop2 to a role in modulating substrate wetting (Ryan et al, 2001; Douezan et al, 2011; Pérez-González et al, 2019), thus acting on the balance of cortical tensions. We should clarify at this point that actin polymerization-based protrusive activity also contributes to cell spreading and is certainly an essential component for cell migration. However, we consider that this aspect is unlikely to account for the EpCAM and Trop2-dependent phenotypes: In MCF7 cells, dynamic protrusions were observed in all situations, and any degree of cell spreading, from flat EpCAM KD cells to round Trop2 KD cells (e.g., Fig. 1I; Appendix Fig. S5). As for

HCC1500 cells, most did not even form lamellipodia on collagen gel, and EpCAM depletion induced spreading of the whole cell "body", generally in the absence of detectable "protrusions", as evident from the z profiles. In addition to the balance between tensions at the free surface and at the matrix interface, EpCAM KD and Trop2 KD also respectively increased or decreased cell-cell adhesion and tissue cohesion. Again, the same trend was found in the three different cell lines.

The opposite EpCAM KD and Trop2 KD phenotypes are reminiscent of the wetting-dewetting transition observed upon induction of increasing levels of E-cadherin in MDA-MB-231 cell aggregates (Pérez-González et al, 2019). The authors concluded that E-cadherin increased the internal tension of the aggregate, thus counteracting substrate wetting. Our analysis of EpCAM and Trop2 KD phenotypes points to a more sophisticated situation: Firstly, there were clear impacts on both cell-matrix and cell-cell interactions. Secondly, EpCAM KD and Trop2 KD, despite causing a similar increase in global contractility and similar recruitment of E-cadherin at contacts, seemed to differentially influence two concurrent processes, i.e., (a) tension-dependent stimulation of mechanical coupling and (b) dampening contractility at contact interfaces relative to free edges. The former appeared preferentially stimulated by Trop2 KD (see "acute" vinculin recruitment), the latter more prominent upon EpCAM KD. It is the integration of these various parameters that accounts for the poor wetting of stiffer Trop2 KD spheroids, and explains how EpCAM-depleted spheroids managed the feat of maintaining high tissue cohesion and, at the same time allowing extensive amount of intratissue cell movements (intercalation) required for their extensive spreading. A similar state is typically observed in compact yet soft and highly migratory embryonic tissues (e.g., David et al, 2014).

This interpretation was fully supported both by our analysis of cellular markers and by direct biophysical measurements. We found that a simple model based on changes in the balance between the three tensions $\gamma_m$, $\gamma_c$, and $\gamma_x$ provides an explanatory logic for all the observed phenotypes. An important feature of this model is that relatively small variations can have a large effect on the final behavioral output. Modulation of this balance can, in turn, be accounted for by partial differential distribution of the two negative regulators, EpCAM comparatively more abundant at free edges and Trop2 at contact interfaces, the bias in Trop2 localization being further exacerbated upon EpCAM depletion. One should emphasize that this deliberately simple and coarse grain model integrates in single tension terms the

huge complexity of the system, including the "anti-adhesive" contractility of the cortex, the "proadhesive" cytoskeletal anchoring of CCC and FAs, and the multitude of regulatory mechanisms acting both a global and local scales. Dissection of these various contributions will be an important but daunting challenge, especially for tissues on soft substrates, where stress fibers, for instance, are largely merged with the cortex.

Comparison of the three cell lines offers a preliminary glance at the spectrum of behaviours that can be achieved by the combined activity of the two partly concurrent, partly antagonistic regulators (Fig. 9). Interestingly, while in all three cell lines, EpCAM was comparatively biased toward edges and Trop2 toward contact interfaces, the overall distribution of the two molecules varied in each line, more at edges in MCF7 cells, more at contacts in HCC1500, intermediate in MDA-MD-453 cells (Figs. 6–8). It is difficult to predict the implications of interfacial tensions, considering that multiple other factors must be involved in setting cortical contractility and adhesion. It is, however, tempting to speculate that EpCAM and Trop2 might both tend to be recruited, each to a different degree, to sites of strong tension, in order to keep contractility at bay. We propose that EpCAM and Trop2 modulate cell behavior without necessarily overriding the intrinsic cell characteristics. Wild-type HCC1500 cells, for instance, spread very poorly on collagen, and, while they did spread slightly better upon EpCAM KD, they were definitely not flattened to the same extent as MCF7 cells. Conversely, in the case of the much less coherent MDA-MD-453 cells, EpCAM KD or Trop2 KD influenced spheroid compaction, but did not significantly affect their fast expansion. EpCAM and Trop2 LOF wetting-dewetting phenotypes were most spectacular in MCF7 cells, presumably because these cells are in a regime which is close to a turning point, where moderate changes in tension can significantly tilt cell and tissue spreading behavior.

Collagen gel were systematically deformed by MCF7 cells and spheroids (Figs. 1 and EV2B), a typical phenomenon of soft substrates. This was characterized in a recent study on S180 sarcoma cell aggregates, where spreading and substrate deformation were described based on an elastocapillary model (Yousafzai et al, 2022). In our images of epithelial MCF7 cells, a capillary effect at the edge of the spheroids was not obvious, perhaps due to different properties of our collagen gels, or differences in cell types. Also, we estimated that the effect of deformation on global spheroid geometry, and thus on the balance of tensions, was small enough to be ignored in our simulations.

The results of the present study challenge some conclusions from previous studies. The increased spreading and cohesion of EpCAM-depleted MCF7 cells may seem at odds with the proadhesive role of EpCAM reported in fish and amphibian embryos and human intestine (Slanchev et al, 2009; Maghzal et al, 2010, 2013; Sivagnanam et al, 2008; Salomon et al, 2017; Barth et al, 2018). One likely difference may lay in the capacity of adhesive structures to cope with increased cortical tension: In the case of Xenopus embryos, cell-cell contacts failed to resist the excessive myosin overactivation caused by EpCAM KD, which resulted in severe stiffening, the collapse of adhesion, and eventually tissue disruption (Maghzal et al, 2013). The same effect probably accounts for the disruption of EpCAM-depleted human intestinal epithelium (Salomon et al, 2017; Barth et al, 2018) In MCF7 cells, the increase in myosin activation and tension experienced upon depletion of either EpCAM or Trop2 were relatively mild, and cells appeared to

be able to compensate through cadherin recruitment. We propose that moderate and balanced EpCAM and Trop2 levels (Appendix Fig. S1) contribute to set favorable conditions for these two regulators to manifest their differential localization and thus balance each other to fine-tune contractility.

Another related unsolved issue is the apparent redundancy during mouse development, which appears normal for single EpCAM or Trop2 knock-outs, and the partial compensation observed in tissues where both are expressed (Lei et al, 2012; Guerra et al, 2012; Wang et al, 2011; Szabo et al, 2022). One piece of explanation may relate to the acute LOFs used in our study (over 2–5 days). It could be conceivable that, in the longer term, cells adapt to this loss by rerouting the remaining partner. We propose that, in the absence of compensatory mechanisms, EpCAM and Trop2 may be interchangeable for their generic action on contractility, but not for a more refined balance of tension required in particular tissues. This is supported in the case of human CTE, where intestinal tuft formation caused by mutations in EpCAM could only be partially rescued by ectopic Trop2 expression (Nakato et al, 2020).

While all our observations amply support the view that cortical actomyosin contractility is the major target of EpCAM and Trop2 as far as cell behavior is concerned, we should not exclude other alternate or complementary possibilities, including differences in the biochemical activities of the two proteins. At the moment, myosin downregulation is the only firmly validated property related to morphogenetic properties, but future studies may provide additional mechanisms. Furthermore, EpCAM and Trop2 may not have the same biochemical activity at all interfaces, but could impinge on different molecular pathways. Such examples exist, such as the specific repression of myosin at cell contacts by the DDR1-Par3/6 complex (Hidalgo-Carcedo et al, 2011). A first hint that different subcellular pools could be at work is the more punctate IF signal for Trop2 and its sensitivity to detergents. Furthermore, the abovementioned differences in the global distribution of EpCAM and Trop2 in different cell lines point to additional levels of regulation. Solving the details of this system, including the molecular basis of differential localization, will not be straightforward: All properties reported for EpCAM appear to be shared by Trop2, including oligomerization, association with tetraspanin-based domains, with claudins, potential interaction with the cytoskeleton, or matriptase-mediated cleavage (Guerra et al, 2022; Szabo et al, 2022; Wu et al, 2020; Fagotto and Aslemarz, 2020; Pavšič, 2021; Balzar et al, 2001; Pavsic et al, 2014; Kuhn et al, 2007; Nubel et al, 2009; Balzar et al, 1998). Both form homodimers, but structural considerations indicate incompatibility for hetero-dimers (Pavšič, 2021). A related puzzling phenomenon is the sensitivity of Trop2 subcellular localization to EpCAM depletion, indicating a complex relationship, probably involving both cooperation and competition for recruitment to sub-compartments and/or for targeting degradative routes.

While we are well aware of the caveats inherent to any LOF/GOF experiments, we consider that the depletion/overexpression approach used in this study is of clear physiological relevance, since the magnitude of the changes in EpCAM and Trop2 levels is compatible with what happens during cancer development, including the abrupt drop in EpCAM expression observed after EMT (Sankpal et al, 2017). While this study has only addressed the function of EpCAM and Trop2 in three cell lines in an in vitro context, we hypothesize that co-expression of EpCAM and Trop2

may act as a general mechanostat contributing to setting the adhesive and migratory properties, which may be broadly relevant in the context of the biology of carcinoma. Both EpCAM and Trop2 are strongly expressed in a large number of cancer cell lines and multiple types of human cancers. Our data may at least partly explain why one cannot assign a general unambiguous pro- or anti-invasive activity to these molecules. Indeed, their action shall not only depend on expression levels, but crucially on their relative levels, and will be further strongly influence by other, cell type-dependent properties, as exemplified in our study. One may also reasonably predict that cancer cells can modulate this EpCAM-Trop2 system, resulting in more plasticity during various phases of cancer development.

# Methods

### Reagents and tools table

**Cell lines**

| Cell line | Origin | Supplier | Cat. # |
|---|---|---|---|
| MCF7 (*H. sapiens*) | Breast Adenocarcinoma | SIRIC Montpellier center, courtesy of Dr. M. Lapierre, originally from ATCC | HTB-22 |
| HCC1500 (*H. sapiens*) | Breast Carcinoma | | CRL-2329 |
| MDA-MB-453 (*H. sapiens*) | Breast Carcinoma | | HTB-131 |

**Antibodies**

| Antigen | Supplier | Antibody name/ cat # | Species | Dilution for IF unless stated for WB |
|---|---|---|---|---|
| EpCAM | Santa Cruz Biotechnologies | 323/A3 | Mouse | 1:1000 |
| Trop2 | Invitrogen/ Thermo Fisher | MA5-29593 | Rabbit | 1:1000 |
| Trop2 | R&D | AF650 | Goat | WB 1:500 |
| pMLC(Ser19) | Cell Signaling | 3675 | Mouse | 1:200 |
| pMLC(Ser19) | Millipore/Merck | AB3381 | Rabbit | 1:1000 |
| E-cadherin | Cell Signaling | 24E10 | Mouse | 1:400 |
| β-catenin | Santa Cruz Biotech | H102/#3195 | Rabbit | 1:250 |
| vinculin | Boster Bio | MA1103 | Mouse | 1:100 |
| α-tubulin | Sigma/Merck | T8203 | Mouse | WB 1:4000 |
| GFP | Aves Labs | GFP-1020 | Chicken | 1:1000 |

**siRNAs**

| siRNA | Supplier | Catalog Number |
|---|---|---|
| control | Santa Cruz Biotechnologies | sc-37007 |
| EpCAM | Santa Cruz Biotechnologies | sc-43032 |
| Trop2 | Santa Cruz Biotechnologies | sc-72392 |
| Trop2 | Sigma/Merck | H502-0033 |

**Recombinant DNA**

**Cell lines**

| Cell line | Origin | Supplier | Cat. # |
|---|---|---|---|
| Gene product | Backbone plasmid | Description | |
| EpCAM-GFP | pSBtet Pur | Human EpCAM, Mutated GC99AT, resistant to siEpCAM, C-term fused with GFP via 5x Gly linker | |
| Trop2-GFP | pSBtet Pur | Human Trop2 fused to GFP term via 5x Gly linker | |

**Various Chemicals and Reagents**

| Reagent | Supplier | Catalog Number | Dilution/ Concentration |
|---|---|---|---|
| Acrylamide | Sigma/Merck | A4058 | 40% |
| Alexa Fluor™ 647 Phalloidin | Molecular Probes/ Invitrogen/ Thermo Fisher | A22287 | 1:2000 |
| Antibiotic-Antimycotic | Gibco/ Thermo Fisher | 15240096 | 1:100 |
| Bind-silane | Sigma/Merck | GE17-1330-01 | |
| Bisacrylamide | Sigma/Merck | M1533 | 2% |
| CellMask deep red | Molecular Probes/ Invitrogen/ Thermo Fisher | C1004 | 1:10000 |
| Dulbecco's modified Eagle's medium (DMEM) high glucose | Thermo Fisher | 11965084 | |
| EdU click reaction kit, Alexa Fluor 488 | Molecular Probes/ Invitrogen/ Thermo Fisher | C10337 | |
| Far-red fluorescent nanobeads | Bangs laboratory | FC02F | |
| Hoechst 33342 | Molecular Probes/ Invitrogen/ Thermo Fisher | H3570 | |
| Immobilon Western reagent | Millipore/Merck | WBKL505500 | |
| Lipofectamine RNAiMAX | Invitrogen/ Thermo Fisher | 13778100 | |
| Methylcellulose | Sigma/Merck | M0262-100G | |
| Mitomycin C | Sigma/Merck | M4287 | 2.5 μM |
| non-essential amino acids | Gibco/ Thermo Fisher | 11140050 | 1:100 |
| Paraformaldehyde | Electron Microscopy Sciences | 15714 | 3.7% |
| Phosphate Buffer Saline wo Ca2+ / Mg2+ | EuroBio | CS1PBS01-01 S36972 | |
| Polylysine-PEG | JenKem Technology | PLL20K-G35-PEG2K | |
| Sheep serum | Biowest | S2350-100 | 20% |
| Trypsin-EDTA | Gibco/ Thermo Fisher | 25300104 | 0.05% |

| Cell lines | | | |
|---|---|---|---|
| Cell line | Origin | Supplier | Cat. # |
| **Cover glasses and dishes** | | | |
| 35 mm Glass-bottom dishes | Cellvis | D35-20-1.5-N | |
| round-bottom 96-well plates | Greiner Bio-One Cellstar, | 650185 | |
| 32 mm round cover glasses N1 | VWR | 631-0162 | |

## Antibodies and reagents

The list is presented in the Reagent and Tool Table.

## Cell culture

MCF7, HCC1500, and MDA-MD-453 cells were grown in complete culture medium, DMEM high glucose, supplemented with 10% fetal bovine serum, 1% antibiotic-antimycotic, and 1% non-essential amino acids, at 37 °C and 5% $CO_2$, for a maximum of ten passages. 0.05% trypsin-EDTA was used for dissociation and passage.

## siRNA transfection

Cells were seeded in 24-well plates at $0.2 \times 10^6$ density one day before siRNA transfection, and 1.6 ml fresh medium was added just before transfection. About 40 pmol siRNA ($2 \times 40$ pmol for dKD) and of 4 µl Lipofectamine RNAiMAX (8 µl for 2KD) were added to each well. Efficient KD was obtained after 48 to 72 h.

## Fibrillar collagen gels

35-mm glass-bottom dishes or 12 mm round coverslips were plasma cleaned to create a hydrophilic surface, then coated at 4 °C with a thin layer of a 2 mg/ml solution of collagen I (from a ~3 mg/ml stock solution in 0.02 N acetic acid) in PBS, titrated to alkalinity by addition of NaOH, and incubated at 37 °C for 30 min for gelling.

## Spheroid formation

The siRNA-transfected cells were dissociated by trypsin and counted using a trypan-blue solution and an automated cell counter (Invitrogen, Countess). For each spheroid, 400 cells were suspended in 200 µl complete culture medium containing 20% methylcellulose and incubated in single wells of round-bottom 96-well plates for 48 h.

## Spheroid migration assay

The formed spheroids were transferred to glass-bottom dishes coated with collagen gel. Adherence to the gel was observed within 30 min. Spheroids were imaged with an Olympus IX83 inverted widefield video-microscope controlled by Metamorph software using a 10x (0.3NA) or 20x (0.45NA) objective. Time lapse images were acquired every 30 min for 24 h.

## Single-cell migration assay

Cells were dissociated in dissociation buffer and counted as above, $0.5 \times 10^5$ cells were plated on top of collage gel and left to adhere for about 3 h before acquisition of time lapses. Imaging conditions were as for spheroids, with a 20x (0.45NA) objective, every 5 to 7 min for 15 h. Migration was measured using the MetaMorph software tracking tool.

## Immunostaining

Spheroids (in glass-bottom dishes) were fixed for 30 min in 3.7% paraformaldehyde in PHEM buffer (60 mM PIPES, 25 mM HEPES, 10 mM EGTA, and 4 mM MgSO4, pH 7.2), followed by 30 min permeabilization in 1% Triton X-100 in PBS. Fixed samples were washed twice with TBS, incubated for 45 min at RT in blocking buffer (20% sheep serum in PBS, followed by overnight at 4 °C with primary antibodies and fluorescent phalloidin diluted in PBS with 10% sheep serum, then another overnight with secondary antibodies and Hoechst 33342, rinsed and kept in PBS. Fixation and immunostaining of cells and groups of cells (on coverslips) were similar, but with shorter times (10 min fixation, 10 min permeabilization, 1–2 h incubation with antibodies, all at RT), and coverslips were mounted using antifade mounting media Slowfade. For cell-surface labeling, Triton X-100 permeabilization after fixation was skipped, but samples were post-fixed after primary antibody incubation, and permeabilized before incubation with the secondary antibodies.

## Confocal microscopy

z-stacks (0.2 to 0.4-µm distance between planes) of immunolabelled cells and groups of cells were acquired using an inverted scanning confocal microscope (Leica SP5-SMD), 63x oil objective (1.4NA). Spheroids were imaged using a Nikon inverted microscope coupled to the Andor Dragonfly spinning disk, 40X water objective (1.15NA) as z-stacks of 4 by 4 tile-scans (1-µm distance between planes), which where stitched to generate full images of whole spheroids. Images from z-stacks were deconvolved using the Huygens software (Scientific Volume Imaging), using the built-in standard template.

## Image analysis

All analyses were performed using Fiji/ImageJ (https://fiji.sc). Experimenters were not blinded to group assignment and outcome assessment.

## Quantification of immunofluorescence images

In all cases, cytoplasmic signal was taken as background, and subtracted from the corresponding intensity values. For measurement of relative E-Cadherin and pMLC signal intensities of whole-mount labeled spheroids, the membrane signal was obtained from horizontal confocal slices through thresholding after background subtraction. Values were normalized for each experiment to the average signal in control spheroids. For small groups of cells, peak intensities for pMLC and E-cadherin were measured by drawing a line across two adjacent cells, perpendicular to the cell edges, and

running the corresponding plot profile function throughout the z stack to select the planes with respectively highest E-cadherin and pMLC peaks at cell-cell contact, and highest pMLC peaks at each free edge. These measurements were repeated for each cell pair of a group. Relative intensities were calculated from the areas of these peaks after background subtraction. Vinculin FA quantification was based on thresholding and measurement of average intensity and FA total surface per cell (or cell group). The integrated intensity was calculated as vinculin average intensity x surface. Normalization to cell size was obtained by dividing by the maximal horizontal cell area. For vinculin-β-catenin double staining experiments, maximal projections were produced from sub-stacks lacking the ventral planes that had signals from FAs. Both β-catenin and vinculin membrane intensities for each field were extracted from a mask created based on the β-catenin signal. EpCAM and Trop2 relative signals in cell groups were measured as average intensity on line scans drawn along the plasma membrane at defined locations, either on horizontal sections or on z projections produced using the "reslice" function. The detailed distributions in MCF7 cells are detailed in Appendix Fig. S8. The same quantification based on manual drawing of lines along cell interfaces was also used for all the data on HCC1500 et MDA-MD-453 cells.

## Quantification of cell morphology. MCF7 spheroids

Confocal z-stacks of spheroids, immunostained for E-cadherin and pMLC or EpCAM and Trop2 were used. The spheroid area was directly measured from the z stack maximal projection, and an average diameter was calculated assuming a circular shape, thus:

$$A = \pi \left(\frac{d}{2}\right)^2 => d = 2\sqrt{\frac{A}{\pi}}$$

The reslice function of ImageJ was used to obtain orthogonal views, which were used to estimate both spheroid height and angles at edges. Spheroids not being fully transparent, the very top of compact spheroids (mostly siTrop2) was often not visible, in which case an approximate height was interpolated based on the shape of the visible portion.

## MCF7 single cells

Dissociated cells laid on collagen gel were stained with phalloidin and Hoechst and imaged by confocal microscopy. Quantification of cell area, cell height, and contact angles were performed for spheroids. Angle measurements were much less accurate, due to the irregular profile shapes and protrusions. For CPM simulations, we used average height/diameter ratios simplified from which we deduced corresponding disk fragments (ratios: 0.5 for controls, 0.37 for siEpCAM, 0.6 for siTrop2, Fig. 5E). We verified that these approximated shapes had edge angles similar to those measured from cross-sections (Appendix Fig. S4).

## HCC1550 single cells and groups

z confocal stacks were resliced to obtain vertical projections from four orientations (diagram in Fig. EV5A', and maximal projections were used to evaluate angles made between the edge and the ventral interface (acute, roughly right, or weakly acute, obtuse). Multiple combinations were compiled into five categories detailed in Fig. EV5.

## Traction force microscopy

Stock solution for soft polyacrylamide substrates of 5 kPa rigidity containing 0.19-µm far-red fluorescent nanobeads were prepared by mixing 40% acrylamide and bisacrylamide 2% in PBS according to documented relative concentrations (Tse and Engler, 2010; Vignaud et al, 2014). The thin polyacrylamide-based substrate was polymerized between two different 32 mm glass coverslips prepared as follows. The first coverslip served as the base for the gel. It was cleaned using a plasma cleaner machine, then coated with bind-silane for 3 to 5 min at RT to ensure the attachment of the gel to the coverslip. The second coverslip served to transfer the patterned extracellular matrix. It was also first plasma cleaned, then coated with 0.1 mg/ml PLL-PEG solution for 30 min at RT to obtain a passivated surface. It was then washed with distilled water, dried, then burned for 5 min with UV light through a micropatterning chrome photomask (45 by 45 µm custom-designed H shapes, micropatterned onto chrome photomask by Toppan). This allowed adsorption of the collagen coating at the burned sites resulting in a micropatterned coated coverslip. Collagen type-I was added at 0.5 mg/ml in 0.02 N acetic acid and left for 45 min at RT. For gel polymerization, 1 µl of 10% ammonium persulfate, 1 µl of TEMED, and 0.35 µl of the abovementioned nanobeads were added to 165 µl of the acrylamide-bisacrylamide stock solution. 47 µl of this solution were used to put between the two coverslips for polymerization (30 min, RT). Once polymerized, the collagen-coated top coverslip was gently removed, exposing the collagen H micropatterned gel. Cells were plated on this substrate at a density of $0.5 \times 10^5$ per coverslip in a culture dish. The medium of each dish was replaced with fresh medium to wash out cells that didn't adhere to the substrate (this step avoided ending up with cells that were not on the patterns). The dishes were then kept in the incubator overnight to allow cell division to obtain cell doublets on each H micropattern. Cells and the underneath nanobeads were imaged using an epifluorescence inverted microscope (Nikon Ti2-E2) with 40x air objective (1.15NA) and an Orca Flash 4.0 sCMOS camera (Hamamatsu), with the temperature set at 37 °C. This first image served as the stressed (pulled) state of the beads. Then, the cells were removed from the patterns using trypsin, and another image of the same position was taken, serving as an unstressed state of the beads. The displacement field analysis was done using a custom-made algorithm based on the combination of particle image velocimetry and single-particle tracking. Traction forces were calculated using Fourier transform traction cytometry with zero-order regularization (Sabass et al, 2008; Milloud et al, 2017). Total contractile energy Ec was calculated based on the formula (Tseng et al, 2012).

$$E_c = \frac{1}{2}\sum_{i,j} F_{i,j} u_{i,j}$$

With $E_c$ being the contractile energy, $i$, $j$ being the pixel positions, $F_{i,j}$ the traction forces at each pixel and $u_{i,j}$ the displacement value at each pixel. All other metrics were calculated as in (Tseng et al, 2012). All calculations and image processing were performed using MATLAB software.

## Production and analysis of free-floating cell doublets

siRNA transfected cells were dissociated by incubating at RT for 10 min in dissociation buffer (88 mM NaCl, 1 mM KCl, and 10 mM NaHCO$_3$, pH = 9.5, Canty et al, 2017), transferred to 1 ml of complete culture medium, and rapid pipetting to yield single cells. Dissociated cells of two different conditions (with control cells pre-labeled with Hoechst 33342) were gently mixed, transferred to agarose-coated plastic dishes (2% agarose in PBS) in complete culture medium and put in the incubator for 10 to 15 min to allow partial reassociation. This time was empirically determined as sufficient to reach close to maximal cell-cell contact expansion while minimizing the formation of larger groups of cells. Re-associated cells were then gently transferred to a glass-bottom dish coated with a thin layer of 2% agarose, in a complete culture medium containing a 1:10,000 dilution of the membrane dye CellMask deep red. Cell doublets were imaged by live confocal microscopy using a Dragonfly spinning disk (Andor) mounted on a Nikon inverted microscope with a 20x (0.75 NA/oil) objective, at 37 °C and 5% CO$_2$. Hoechst and CellMask images were obtained simultaneously using two CCD cameras (EMCCD iXon888 Life Andor). Quantifications of the relative cortical tensions and adhesiveness were done based geometry of cell membranes at the vertices, as previously described (Kashkooli et al, 2021). Control cells in homo- and heterotypic doublets were identified based on Hoechst-positive nuclei.

## Western blot

MCF7 cells were collected in lysis buffer (270 mM sucrose, 50 mM Tris-HCl pH 7.5, 1 mM EDTA, 50 mM sodium fluoride, 5 mM sodium pyrophosphate, 1 mM sodium orthovanadate, 0.5% Triton X-100, 1 mM Benzamidine, 1 mM 4-(2-aminoethyl)benzenesulfonyl fluoride hydrochloride, 1 mM phenylmethylsulfonyl fluoride). They were analyzed by standard SDS-PAGE, using 12% acrylamide gels, and transferred to nitrocellulose. For EpCAM detection, SDS-PAGE was performed in the absence of a reducing agent. Membranes were blocked in 5% milk, in TBS-T buffer (50 mM Tri-HCl, 150 mM NaCl, 0.1% Tween, pH 7.5). Antibodies were diluted either in 1% milk (EpCAM, α-tubulin) in TBS-T, or in 10% sheep serum, 3% bovine serum albumin in TBS-T (Trop2). Secondary antibodies were peroxidase coupled, and chemiluminescence signal was imaged with an Amersham Imager 600, and quantified using the ImageJ gel analysis tool. Linearity of signal was verified by loading samples dilutions. For re-blotting, membranes were stripped for 3x 10 min in 1.5% glycine, 0.1% SDS, 1% Tween, pH 2.2.

## Statistical analysis

All statistical analyses were performed using the Excel real statistics add-in. The numbers of samples and of independent experiments are indicated in the figures, above the graphs. Comparisons between multiple conditions were done using either one-way ANOVA followed by Tukey-HSD post hoc test, or, in case of variance not similar between conditions or low sample number(<10), the non-parametric Kruskal–Wallis test, followed by Dunn post hoc test.

## Data availability

The source data for this study have been uploaded to BioStudies: https://www.ebi.ac.uk/biostudies/studies/S-BSST1685.

The source data of this paper are collected in the following database record: biostudies:S-SCDT-10_1038-S44318-024-00309-9.

## Peer review information

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

## Acknowledgements

We acknowledge the help of the MRI imaging platform. We thank Marion Lapierre (IRCM Montpellier) for advice and for providing cell lines. We thank Rudi Winklbauer for the critical reading of the manuscript and very valuable advice. This study was supported by ANR (Agence Nationale de la Recherche) grant ANR-14-ACHN-0004–ICM, ANR-21-CE13-0042-01, and a Labex EpiGenMed Chair of excellence ANR grant to FF, and Canadian Institute of Health Research grant 130350 to FF and PL.

## Author contributions

**Azam Aslemarz**: Conceptualization; Formal analysis; Investigation; Methodology; Writing—original draft. **Marie Fagotto-Kaufmann**: Formal analysis; Investigation; Methodology; Writing—original draft. **Artur Ruppel**: Formal analysis; Investigation; Methodology. **Christine Fagotto-Kaufmann**: Formal analysis; Investigation; Methodology. **Martial Balland**: Supervision. **Paul Lasko**: Supervision; Funding acquisition. **François Fagotto**: Conceptualization; Formal analysis; Supervision; Funding acquisition; Investigation; Methodology; Writing—original draft; Writing—review and editing.

Source data underlying figure panels in this paper may have individual authorship assigned. Where available, figure panel/source data authorship is listed in the following database record: biostudies:S-SCDT-10_1038-S44318-024-00309-9.

## Disclosure and competing interests statement

The authors declare no competing interests.

# Expanded View Figures

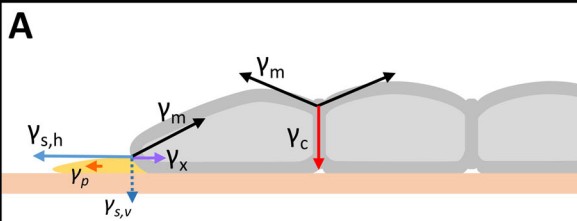
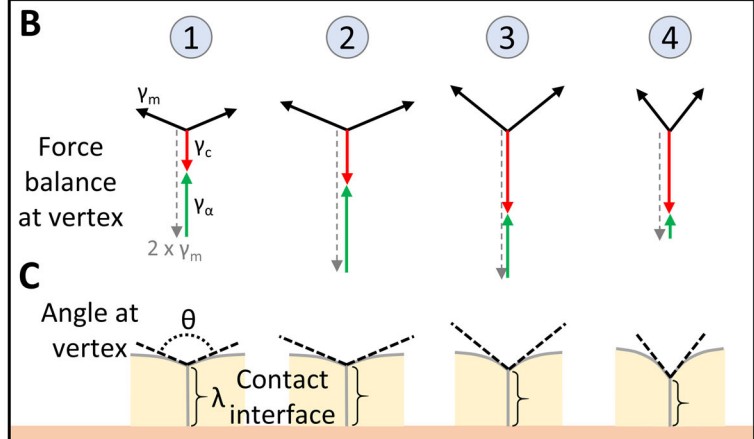

**Figure EV1. Summary of the biophysical description of cell spreading and adhesion based on the balance of tensions at interfaces.**

(A) The capacity of cells to spread on a substrate, be it matrix or other cells, is strongly dependent on the contractility of the actomyosin cortex, which can be viewed as analogous to the role of surface tension in the physical process of a liquid wetting a surface (Brodland, 2002; Douezan et al, 2011; Amack and Manning, 2012; Winklbauer, 2015). The system can similarly be described as the balance of the tensions exerted at various interfaces, namely tension at the matrix ($\gamma_x$), at cell contact ($\gamma_c$), and along the free edges exposed to the medium ($\gamma_m$). $\gamma_x$ and $\gamma_c$ combine various tensions exerted at the corresponding interface. In the case of cell-cell adhesion, for instance, $\gamma_c$ is the sum of the cortical tensions on each side of the contact minus the so-called "adhesive tension" (see below). For adhesion to matrix, $\gamma_x$ and $\gamma_m$ are balanced by substrate tension $\gamma_s$. Note that it is customary to consider the horizontal components $\gamma_{s,h}$, which is typically the one measured by techniques such as traction force microscopy. Spreading of cell aggregates is controlled by the combination of cell-matrix and cell-cell adhesion, thus by all three tensions (Ryan et al, 2001; Douezan et al, 2011). Actin polymerization within the advancing protrusion also generates a force $\gamma_p$, which may contribute to the force balance, depending on the degree of the mechanical coupling with the rest of the cytoskeleton. (B) Cells and tissues constitute active materials, and their behavior is clearly more complex than the wetting/adhesion of a classical liquid. Taking the case of cell-cell adhesion, expansion of an adhesive contact $\lambda$ requires that contact tension $\gamma_c$ is lowered compared to the basal cortical tension $\gamma_m$ that would otherwise act on each side of the interface (2 x $\gamma_m$). This is achieved partly by the adhesion tension resulting from the binding of adhesion molecules to their ligands (cadherin-cadherin and integrins-matrix), but mainly through an active downregulation of local actomyosin contractility along the contact interface. This latter process relies on the ability of adhesion molecules to recruit regulators of cytoskeleton remodeling. To highlight this downregulation, we introduce here a single tension vector $\gamma_\alpha$ that includes the ensemble of contributions that decrease $\gamma_c$ (thus 2$\gamma_m = \gamma_c - \gamma_\alpha$). Cadherin coupling to the actomyosin network (Leckband and de Rooij, 2014; Charras and Yap, 2018), and stress fibers anchoring focal adhesions for matrix adhesion, are important additional inputs that impact tensions at the various interfaces. (C) Cell geometry reflects the force balance: As shown in panel (1), a low $\gamma_c/\gamma_m$ ratio (high $\gamma_\alpha$) leads to a large contact interface $\lambda$, and a wide-angle $\theta$. The latter is a direct geometric readout of "adhesiveness" (David et al, 2014). A corollary is that a smooth surface directly reflects the high cohesivity of a tissue (Amack and Manning, 2012; Winklbauer, 2015). (2–4) Contacts adapt to stress through a process of reinforcement, involving the recruitment of cortical cytoskeleton and of adhesion molecules, and increased linkage of adhesion molecules to the cytoskeleton (Engl et al, 2014; Charras and Yap, 2018). As shown in panel (2), cells can then maintain the same degree of adhesiveness (same angle $\theta$) despite bearing higher tensions. Importantly, this also requires more repression of contractility along the cell-cell interface (increased $\gamma_\alpha$), another expected effect of enhanced cadherin recruitment. Panel (3) shows the situation where higher cortical tension is not compensated by increased $\gamma_\alpha$, resulting in a higher $\gamma_c/\gamma_m$ ratio and lower adhesiveness (smaller angle $\theta$). In the last example (4), $\gamma_m$ remains unchanged compared to (1), but $\gamma_\alpha$ is decreased, resulting in relatively high $\gamma_c$, and thus low adhesiveness (low angle $\theta$ and shorter contact interface $\lambda$. The same principles are applicable to adhesion and spreading to the extracellular matrix, which depends on the balance between $\gamma_x$ and $\gamma_m$. Here, in addition to cortical contractility, the contribution of stress fibers also impacts the balance between $\gamma_x$ and $\gamma_m$.

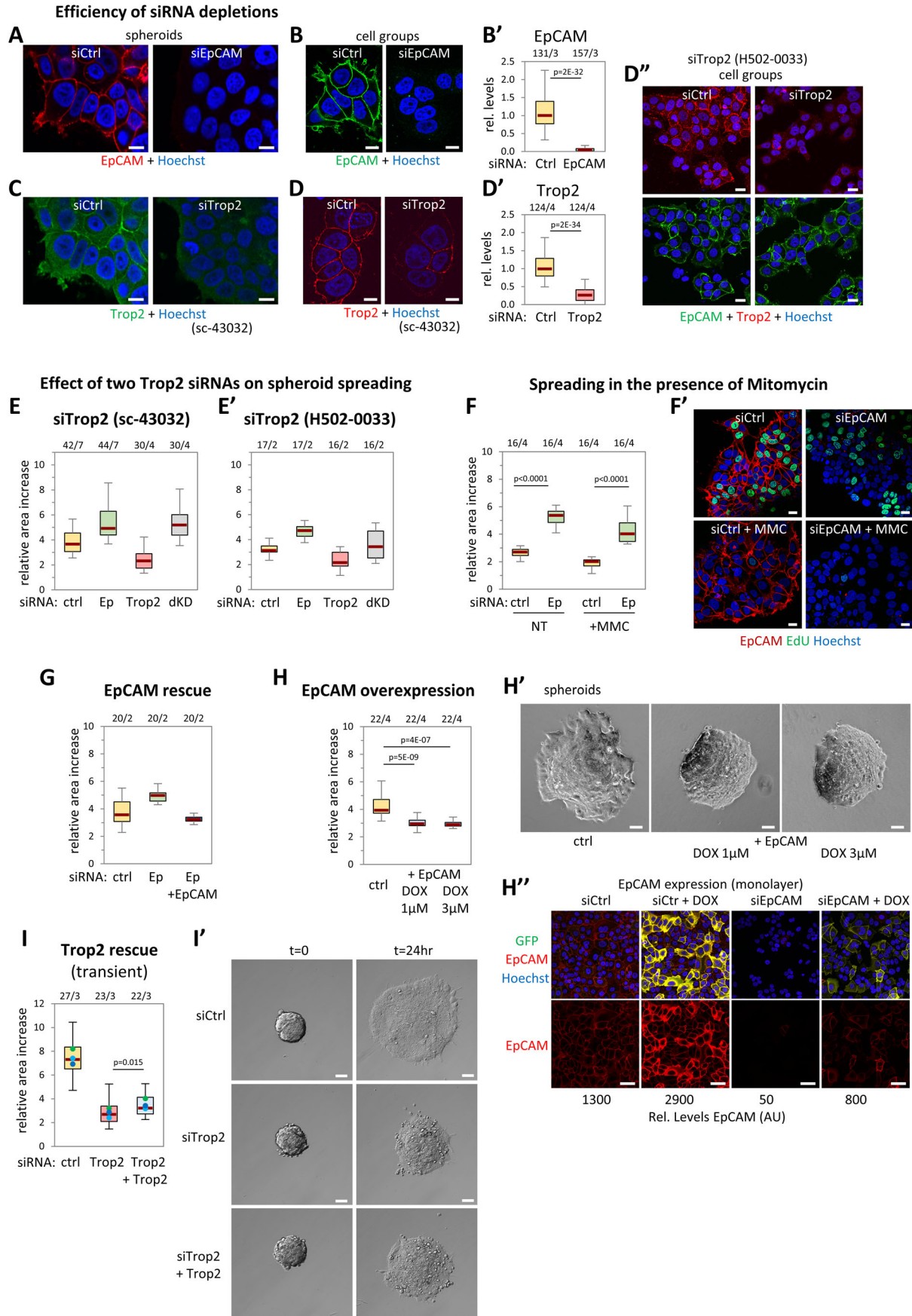

**Figure EV2. EpCAM and Trop2 depletions, rescues, and other controls.**

(A–D) EpCAM and Trop2 depletions. (A, C) Representative confocal microscopy images of the edge of spheroids formed with cell transfected for 96 h with Ctrl, EpCAM, and Trop2 siRNA (sc-43032), immunolabelled for EpCAM and Trop2. Nuclei were stained with Hoechst (blue). The specific signal along cell membranes is undetectable in the respective siRNA condition. (B, D) Confocal microscopy images of groups of cells transfected for 96 h with Ctrl, EpCAM, and Trop2 siRNA (sc-43032), immunolabelled for EpCAM and Trop2. Scale bars: 10 μm. (B', D') Quantification of EpCAM and Trop2 signal intensity at cell-cell contacts. Levels were normalized to the median value for siCtrl cells. The box plots show the interquartile range (box limits), median (center line), and min and max values without outliers (whiskers). Numbers of individual contacts/biological replicates indicated above graphs. Statistical analysis, two-tailed Student's $t$-test. (D") Same staining as in (D), for cells transfected with siCtrl and siTrop2 (H502-0033). Scale bars: 20 μm. (E) Comparison of the effect of siTrop2 sc-43032 and H502-0033 on spheroid spreading. Numbers of spheroids/biological replicates indicated above graphs (in Source Data, (E) = exp1–3,6–9, (E') = exp4,5). Statistical analysis: one-way ANOVA followed by Tukey-HSD post hoc test. (E"). Immunofluorescence for EpCAM and Trop2 of groups of control and H502-0033-transfected cells. (F) Increased spheroid migration upon EpCAM KD is independent of cell proliferation. Spheroids were treated with 2.5 μM mitomycin C (MMC) during the entire migration assay. Numbers of spheroids/biological replicates indicated above graphs. Statistical analysis: one-way ANOVA followed by Tukey-HSD post hoc test. (F') Validation of mitomycin MMC efficiency by imaging EdU incorporation. At the end of the migration assay, the spheroids were incubated for 1 h with thymidine analog EdU, which efficiently incorporates into newly synthesized DNA. EdU was detected in green (see Material and Methods), while EpCAM was detected by immunofluorescence (red), and nuclei were stained with Hoechst (blue). The four panels show representative confocal microscopy images of non-treated, and MMC-treated spheroids of siCtrl and siEpCAM conditions. Maximal projections of 3 z planes, 0.5 m apart. Scale bars: 20 μm. (G) Rescue of EpCAM KD spheroid phenotype. Rescue of spheroid spreading phenotype (24 h) was performed using a mixed population of MCF7 cells stably transfected with a doxycycline (DOX)-inducible EpCAM-GFP variant that included two conservative point mutations within the siRNA target sequence. Ctrl, transfected with siCtrl, no DOX; siEpCAM, no DOX; siEpCAM + 1 μM DOX. Under these conditions, control and siEpCAM spheroids spread to a similar extend as, respectively, regular control and EpCAM KD spheroids (Fig. 1A). DOX treatment fully rescued spreading to control levels. Results from 20 spheroids per conditions, two independent experiments. (H) EpCAM overexpression inhibits spheroid spreading. Spheroids of cells expressing doxycycline-inducible EpCAM-GFP were let spreading for 24 h on collagen gel. Conditions included non-treated control spheroids, and spheroids treated with 1 μM and 3 μM DOX. Results of 22 spheroids, from four independent experiments. Statistical analysis: one-way ANOVA followed by Tukey-HSD post hoc test. (H') Representative examples for the three conditions, 24 h spreading. Scale bars, 50 μm. (H") Representative EpCAM immunofluorescence images of control, DOX-induced, EpCAM-depleted (siEpCAM) and rescued (siEpCAM + DOX) cells. Cells were double labeled for EpCAM, GFP (to detect exogenous, DOX-induced EpCAM). Total relative EpCAM levels in each image are indicated on the right. Scale bars, 50 μm. (I) Transient expression of Trop2-GFP partially rescues the Trop2 KD spheroid phenotype. Spheroids of MCF7 cells transfected consecutively with siTrop2 and a DOX-inducible Trop2-GFP construct, were laid on collagen gel and left to spread in the absence or in the presence of 0.5–3 μM DOX. Positive controls were transfected with siCtrl. Note that spheroids were much smaller than in the other experiments, being formed with 100 rather than 400 cells, and expanded more extensively (~7 folds versus ~4 folds). The reason for this protocol modification was that larger spheroids were more severely damaged by cell death caused by the transient transfection of the Trop2-GFP plasmid, independently of DOX induction. Results from 22–27 spheroids from three independent experiments. Statistical comparison, Student's $t$-test, two-tailed. (I') Examples of control, Trop2 KD (siTrop2 + Trop2-GFP without DOX) and rescue ((siTrop2 + Trop2-GFP with 3 μM DOX). Scale bars, 50 μm.

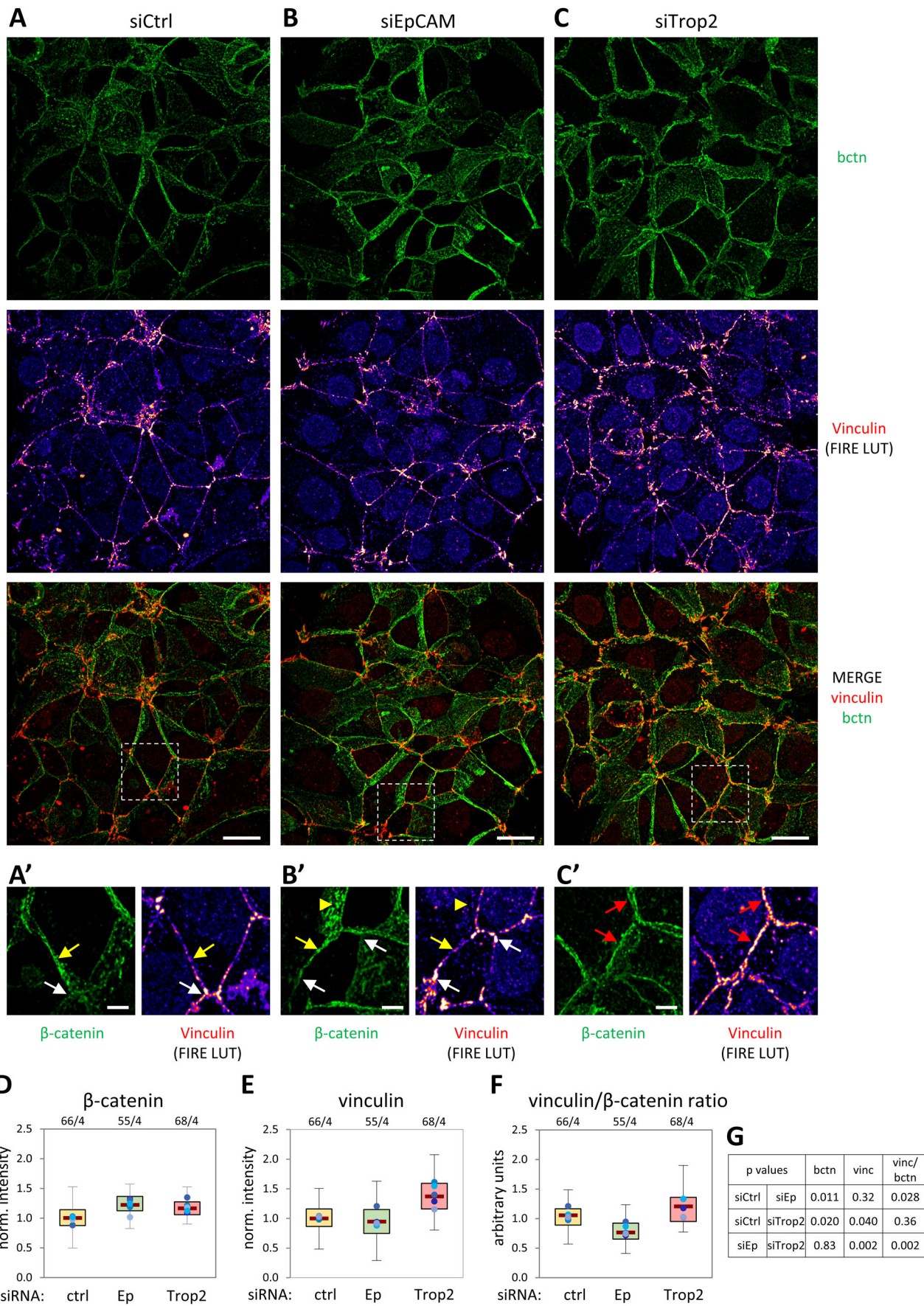

◀ **Figure EV3. Comparison of β-catenin (bctn) and vinculin at cell-cell contacts.**

MCF7 cells were laid on a thin layer of collagen. (**A–C**). Maximal projections from confocal z-stacks, excluding the ventral planes containing the FAs. (**A′–C′**). Enlarged fields from A-C. White arrows, membrane regions with high vinculin and low bctn. Yellow arrows, membrane regions with low vinculin and high bctn. Red arrows in (**C′**), membrane regions with both high vinculin and high bctn. Scale bars: (**A–C**), 20 μm; (**A′–C′**), 5 μm. (**D–G**). Quantification of (**D**) bctn, (**E**) vinculin, and (**F**) vinculin/bctn ratio. The box plots show the interquartile range (box limits), median (center line), and min and max values without outliers (whiskers). Numbers of fields/biological replicates indicated above graphs. One-way non-parametric ANOVA (Kruskal–Wallis Test) on experiment averages, followed by Dunn post hoc test. $p$ values are given in (**G**).

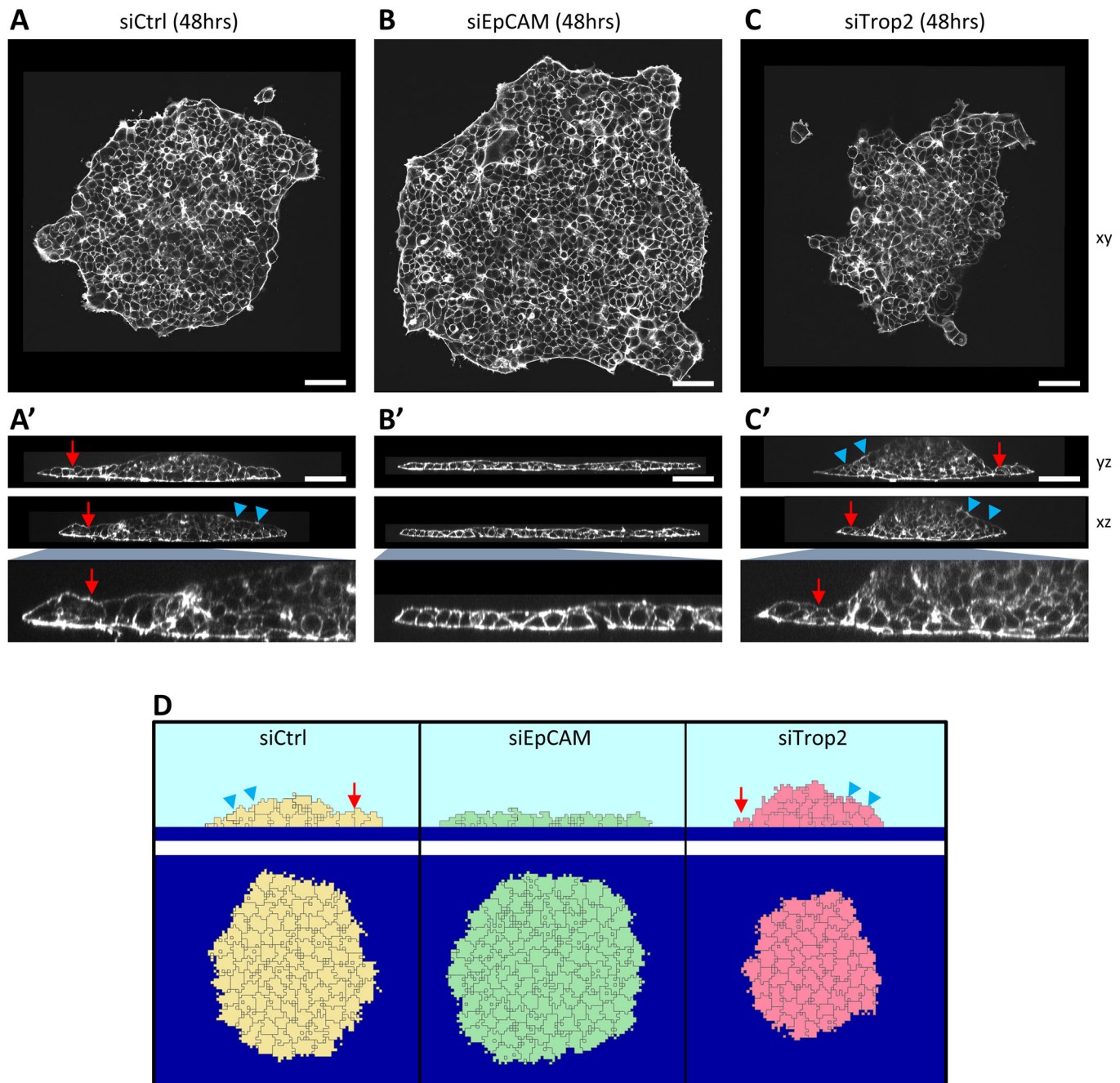

**Figure EV4. Spreading of MCF7 spheroids after 48 h and corresponding simulation.**

(**A–C**) Confocal images from spheroids fixed and labeled with phalloidin. Images are representative of 16 spheroids per condition, from four experiments. (**A–C**) Single horizontal planes correspond to the maximal area. (**A′–C′**) Vertical projections (yz and xz) and enlargement of xz. All spheroids have spread further and thinned compared to 24 h (Fig. 1). EpCAM KD spheroids have reached maximal extension, forming a highly coherent monolayer. All tissues have remained largely compact as indicated from the smooth dorsal surface in vertical projections, although in controls and Trop2 KD, portions tend to protrude (red arrows, smooth portions marked by blue arrowheads). Scale bars: 50 µm. (**D**) Projections from CompuCell3D simulation after 600 iterations (i.e., twice longer than in Fig. 5). The actual morphology of the spheroids is very well recapitulated, including the monolayer for EpCAM KD and the irregular profile for control and siTrop2 conditions (compare red arrows and blue arrowheads).

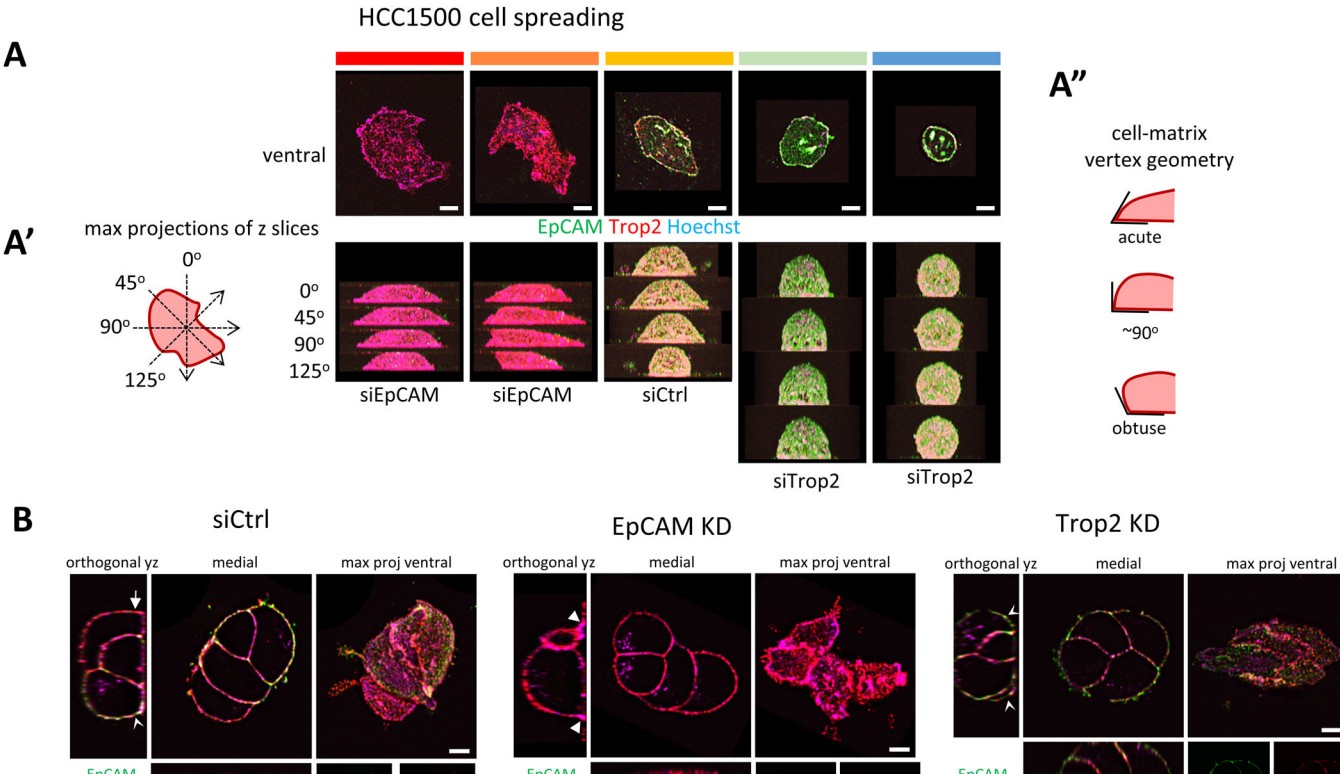

| Category | | Score | Combined edge geometries from four profiles (8 edges) |
|---|---|---|---|
| Spread | Full | 6 | All acute |
| | High | 4 | Most acute |
| | Moderate | 2 | All ~90° or weakly acute<br>Half ~90°, half obtuse<br>Part obtuse, part acute<br>Half acute, half ~90° |
| | Weak | 1 | Most obtuse, rest ~90° or weakly acute |
| Round | | 0 | All obtuse |

| P values | SC | Doublets | Groups |
|---|---|---|---|
| siK-siEpCAM | 0.0003 | 0.35 | 0.006 |
| siK-siTrop2 | 0.33 | 0.022 | 0.93 |
| siEpCAM-siTrop2 | 7E-06 | 0.002 | 0.004 |

**Figure EV5.   Analysis of morphology phenotypes of HCC1500 cells.**

HCC1500 cells transfected with siCtrl, siEpCAM, or siTrop2, laid on fibrillar collagen, were surface immunolabelled for EpCAM, Trop2 and stained for F-actin (Phalloidin). (**A**) Representative examples of single cells, selected to cover the range of degree of spreading. Scale bars, 5 μm. (**A'**) Orthogonal maximal projections in four orientations. (**A"**) These projections were used to observe the geometry at the contact with the matrix substrate, specifically the angle formed by the free cell edge and the substrate interface. (**B**) Examples of siCtrl, siEpCAM, and siTrop2 groups of cells. Each multiple panel includes merged images of a medial horizontal plane, a maximal projection of bottom (ventral) planes, and two orthogonal views xz and xy. Arrowheads point to the free-edge substrate vertex. Filled arrowhead: acute angle; arrow: right angle; convex arrowhead: obtuse angle. For the Trop2 KD example, two xz slices are shown, one with obtuse acute angles (blue arrowhead), one with acute angles (blue arrow). Scale bars, 5 μm. Separate EpCAM and Trop2 channels of the median plane are shown as small inserts for visualization of their respective levels under control and depletion conditions. (**C**) Morphological classification. Single cells and cell groups were classified into five categories of degree of spreading, from round to fully spread, based on the geometry at the substrate vertex. Because of the highly irregular shapes, multiple combinations were pooled in the intermediate categories. (**D**) Statistical analysis corresponding to the results shown in the main Fig. 7C. *P* values from ANOVA analysis followed by Tukey-HSD post hoc test, obtained by allocating to each category a score (0,1,2,4,6). Results were essentially the same using different scales (e.g., 0, 1, 2, 3, 4).

