## [Peer Review File · The EMBO Journal]

An EpCAM/Trop2 mechanostat differentially regulates collective behaviour of human carcinoma cells

Azam Aslemarz, Marie Fagotto-Kaufmann, Artur Ruppel, Christine Fagotto-Kaufmann, Martial Balland, Paul Lasko, and Francois Fagotto

Corresponding author: Francois Fagotto (francois.fagotto@crbm.cnrs.fr)

Review Timeline:

Submission Date:	8th Apr 24
Editorial Decision:	12th Jun 24
Revision Received:	24th Sep 24
Editorial Decision:	2nd Oct 24
Revision Received:	9th Oct 24
Editorial Decision:	23rd Oct 24
Revision Received:	25th Oct 24
Accepted:	25th Oct 24

Editor: Ieva Gailite

Transaction Report:

Dear Dr. Fagotto,

Thank you for submitting your manuscript for consideration by the EMBO Journal. We have now received comments from two reviewers, which are included below for your information.

Based on the overall interest expressed in the referee reports, I would like to invite you to address the comments of both reviewers in a revised version of the manuscript. I should add that it is The EMBO Journal policy to allow only a single major round of revision and that it is therefore important to resolve the main concerns at this stage.

We generally allow three months as standard revision time, which can be extended if necessary. Should you foresee a problem in meeting this deadline, please let me know in advance to discuss an extension. As a matter of policy, competing manuscripts published during this period will not negatively impact on our assessment of the conceptual advance presented by your study. However, please contact me as soon as possible upon publication of any related work to discuss the appropriate course of action.

When preparing your letter of response to the referees' comments, please bear in mind that this will form part of the Review Process File and will therefore be available online to the community. For more details on our Transparent Editorial Process, please visit our website: <https://www.embopress.org/page/journal/14602075/authorguide#transparentprocess>. Please also see the attached instructions for further guidelines on preparation of the revised manuscript.

Please feel free to contact me if you have any further questions regarding the revision. Thank you for the opportunity to consider your work for publication. I look forward to receiving your revised manuscript.

With best regards,

leva

leva Gailite, PhD
Senior Scientific Editor
The EMBO Journal
Meyerohofstrasse 1
D-69117 Heidelberg
Tel: +4962218891309
i.gailite@embojournal.org

We realize that it is difficult to revise to a specific deadline. In the interest of protecting the conceptual advance provided by the work, we recommend a revision within 3 months (10th Sep 2024). Please discuss the revision progress ahead of this time with the editor if you require more time to complete the revisions.

Referee #1:

The manuscript presented by Azam Aslemarz et al. propose a complementary role of EpCam and Trop2 at regulating cell-cell and cell-substrate adhesion/contraction. Depleting EpCAM favor spheroid spreading and favor cell leading edge contractility. Depleting Trop2 has an opposite effect, decreasing spheroid spreading and relocating contractility toward the cell-cell interface. The experimental assays presented are supporting the conclusions and the paper is properly organized and do not require extra experiments or strong reorganization.

The only major concern is the writing that require a careful revision to make the article much more accessible and readable. The author have the tendency to write in an non-informative way that leads to many meaningless paragraphs that are not very useful, maintaining the reader without clear information. For example, at the end of the introduction, the last paragraph said that you have "counter balancing contribution to adhesion and collective migration", or "differential localization and antagonistic effect on cell behavior". This type of writing while pointing to a difference leave the reader without a clue on the message and what each molecule is contributing to. This is present in many places in the article. The entire start of the discussion in particular has the same problem. We have "opposite effect" or "diametrically opposed phenotype" but no clue on who is doing what and where. All the "suspense paragraph" or unclear transitions should be also eliminated. This is very important for the clarity. Readers can have different level of attention and comprehension and repeating much more directly the take home message and the results will help tremendously for the impact of this nice work.

Referee #2:

Review for Aslemarz et al "An EpCAM/Trop2 mechanostat differentially regulates collective behaviour of human carcinoma cells"

Summary

In this manuscript, Aslemarz et al investigate how single or double knock-down of EpCAM and its paralog Trop2 impacts cell and tissue wetting across scales. First, knock-downs are performed in MCF7 cells, and the effective spreading area and solidity (circularity) of spheroids and single cells are determined. Through this, the authors find that EpCAM KD causes increased spreading ("wetting"), while Trop2 KD causes reduced spreading. Next, the authors investigate how KD impacts phospho-myosin2 status, observing that EpCAM KD leads to increased peripheral pMLC, while Trop2 leads to increased pMLC at inner junctions. The authors then examine Vinculin accumulations at cell-substrate and cell-cell adhesions, observing that Trop2 KD appears to increase relative vinculin levels at both adhesion types. The authors then connect this to measurements of force generation at cell-substrate adhesions using traction force microscopy, finding that both EpCAM and/or Trop2 KD lead to increased contractile energy/stress magnitude. They then infer relative cortical tension values and adhesiveness in each KD condition by measuring contact angles between heterotypic cell doublets. Using these experimental data as inputs, the authors then use a cellular Potts model to perform simulations of cellular aggregate spreading events in the different KD conditions, which match the experimentally observed phenotypes. The authors then investigate asymmetries in EpCAM and Trop2 localization patterns across different cellular interfaces, finding that EpCAM has a higher dorsal/ventral membrane localization, while Trop2 has higher relative localization at inner cell-cell junctions. Finally, the authors perform EpCAM and Trop2 KD in two additional cell lines and find consistent results in terms of spreading behavior in single cells and spheroids.

Overall, the manuscript does a nice job of characterizing the phenotypes between EpCAM and Trop2 KDs. The overall finding that knock-down of these two paralogs is quite surprising, and the data presented provides a nice explanation of why this is the case. The authors generally take a quantitative approach to describe the cellular and tissue-level behaviors and using mostly qualitative arguments attribute these changes in wetting behavior to changes in active tensions at cellular interfaces. This is also

nicely complemented with simulations using a cellular Potts model that mimics the experimental observations. There are still a number of aspects of this manuscript that could be improved prior to acceptance for publication, which are listed in a point-by-point manner below:

Major Points

1. It is not clear whether Box 1 is really necessary in the main text. This theory, mostly based on the Young-Dupré equation, has been described in many previous publications applying this to biological systems, albeit with slightly different notation (e.g. Winklbauer, 2015; Maître et al, 2012; Turlier and Maître, 2015). Sometimes this is not always clear from the text, and more effort could be given to discussing this in the context of previous publications, especially in developmental biology, where this theory has been applied extensively, and also in more recent publications using in vitro systems of cell aggregates undergoing active wetting on deformable surfaces (e.g. Pérez-González et al, 2019, Nat. Phys.; Yousafzai et al, 2022, PRX). Also, the theory is mostly used to make qualitative arguments throughout the paper, as tensions are not measured directly. While I do believe that this framework allows for a better understanding of the phenotypes, as it is not used to make quantitative measurements for much of the paper, I am not sure how much focus should be placed on the theory in this case.
2. As EpCAM and Trop2 are paralogs and have potentially overlapping functions, it is possible that there are some regulatory mechanisms to compensate for expression levels. The authors also address this in the Discussion section. It would therefore be important and should be possible to show that expression levels are independent. Ideally this could be done by Western Blot.
3. In Fig. 2, it is shown that Trop2 KD leads to increased pMLC accumulation at internal cell-cell junctions compared to at the cluster edge. In a previous publication, reduction of pMLC at the internal junctions of similarly sized cell clusters on collagen networks was shown to be regulated by DDR1, Par3/6, RhoE, and ROCK (Hidalgo-Carcedo et al, 2011, NCB). It would be insightful to determine whether Trop2 is associated with any of these components and to discuss the results in the context of this previous publication.
4. One of the most striking changes following Trop2 KD is the upregulation of Vinculin at cell-cell junctions. As the effects of Trop2 KD depend on changes in myosin contractility and cell-cell adhesion, increased vinculin recruitment to junctions seems to be an important observation and could be indicative of increased tension at junctions (le Duc et al 2010, JCB; Leckband and de Rooij, 2014 [already cited in manuscript]). Although this seems to be more striking than the difference in vinculin accumulation at FAs, this is never really followed up. Instead, the focus is directed toward stresses at cell-substrate adhesions. Additional investigation of mechanosensing at cell-cell junctions would help to understand this mechanism.
5. The Vinculin stainings and TFM data seems somewhat inconsistent. On one hand, vinculin accumulation at FAs seems to be highest for the Trop2 KD. On the other hand, total traction energy and traction stresses seem to be highest for EpCAM KD, where no change in Vinculin at FAs was observed. As FA assembly is mechanosensitive and higher tractions are typically associated with more mature, FAs, how can this difference be reconciled? And what is meant by the following "adhesion reinforcement in Trop2 KD cells was less efficient" (p. 9)?
6. The TFM data is mostly used to argue that pMLC activity and contractility are higher for both KD conditions. However, the actomyosin structures on the "ventral" surface where the FAs are located is typically very different from the actomyosin cortex on the "dorsal" surface. This should be treated more carefully in the text, as it is not clear how increased tension on FAs relates to cortical tension (γ_m). Also, one is left wondering whether any information related to the cell-matrix tension can be deduced from the TFM experiments. Do the measured traction stresses/energy relate to interfacial tensions in any direct way? It would be helpful to directly address this in the text.
7. Several times throughout the manuscript, the authors reference expression data from public repositories (e.g. Expression Atlas, p. 4, 16; Protein Atlas, p. 12, 16). If this expression data is available, would it be possible to simply generate some plots based on this meta-analysis? This would be more informative for readers than simply citing the repositories.

Minor points

1. The contact angle at the edge of the spreading spheroids is important for understanding spreading mechanics, and the authors include this in Fig. 5. Could this also be measured in the other figures (e.g. Fig. 1H', Fig. 6A",B",C")? It is also unclear why the authors chose to bin the contact angles and define a new "round/spread" scale in Fig. 7C. Although this scale is described well in Fig. S5C, it seems slightly arbitrary and would probably be more straightforward to measure an average (mean or median) contact angle.
2. It is unclear what is presented in Fig. 1G. Are these outlines from actual data, or simply cartoons? I do not think these panels are necessary. Maybe this idea could be better presented by simply outlining the cell aggregates in H/H' for example with a dotted line.
3. The difference in height is striking between conditions presented in Fig. 1 H'. Could this be quantified in addition to the area increase and solidity? Even better would be to measure contact angles.

4. In Fig. 1I', the normalization is slightly unconventional. Height/Area would give a unit of 1/length (e.g. μm^{-1}). No unit is given on the plot. Perhaps simply an aspect ratio from the xz projection (which would be unit-less) would be better here and/or the contact angle at the cell edge.

5. In the quantifications of the dorsal-ventral localizations of EpCAM and Trop2 (Fig. 6E, S6C), there is a lower D-V gradient for Trop2 in both MCF7 cells and HCC1500 cells, which the authors point out in the text. However, the gradient is in the opposite direction for the two cell lines. For MCF7 cells, the ratio is >1 , meaning that both EpCAM and Trop2 are enriched on the dorsal side, while for HCC1500 cells, the ratio is <1 , meaning that EpCAM and Trop2 are enriched on the ventral side. This could be made clearer in the text, and it would be helpful to discuss the implications for the interfacial tensions.

We thank the reviewers for their very helpful comments.

We summarize here the major changes, followed by the point-by-point response.

We have thoroughly corrected the body text. We have in particular reorganized and rewritten sections of the discussion, in order to incorporate points suggested by the reviewers, and hopefully improve its clarity and coherence.

The revised manuscript has highlighted in blue the major parts that have been rewritten or added.

We realized that one key point needed to be clearly stated and explained, i.e. the involvement of two mechanisms acting at adhesive interfaces, i.e. a) the reinforcement of adhesive structures through mechanical coupling with the cytoskeleton and b) the downregulation of cortical tension along the interface, which is a major factor increasing adhesion. We hope that this aspect has been clarified.

Following the editor suggestion, we have moved to the main figures data that previously presented as supplementary data. In particular, we have now two separate figures dedicated for the data on HCC1500 and MDA-MD-453 cells.

We have added two new figures, now part of the “expanded view figures”:

- Based on reviewer 2 suggestion, Fig EV3 presents a new more detailed analysis of cell-cell contacts, performed through quantification of double β -catenin and vinculin staining. This comparison highlights the distinct “states” of the contacts in EpCAM KD, Trop2 KD, consistent with higher adhesiveness for the former, higher tension for the latter.
- Fig. EV4 presents images of late spreading spheroids (48hrs, as opposed to 24hrs used in all other experiments), and a corresponding Cellular Potts Model simulation. This figure has been added because the simulation reproduced remarkably well the new configurations, which we think further validates our model.

As detailed below, we have included some new data and quantifications as suggested by the reviewers.

We have also included some supporting material in the Appendix:

Fig. S5. Details of pMLC and phalloidin staining of spheroids (Fig S5) in order to provide a more complete the description of the cytoskeleton organization of spheroids on soft substrate and effect of EpCAM and Trop2 KD.

Fig. S6. Quantification of phalloidin signal in cell groups, to complement data from Figure 2 (pMLC and E-cadherin).

Please find at the bottom of the point by point response a list of correspondence between figures of the original manuscript and the new figures, that should help to track changes between the two versions.

Point-by-point response

Our answers are in blue

Referee #1:

The manuscript presented by Azam Aslemarz et al. propose a complementary role of EpCam and Trop2 at regulating cell-cell and cell-substrate adhesion/contraction. Depleting EpCAM favor spheroid spreading and favor cell leading edge contractility. Depleting Trop2 has an opposite effect, decreasing spheroid spreading and relocating contractility toward the cell-cell interface. The experimental assays presented are supporting the conclusions and the paper is properly organized and do not require extra experiments or strong reorganization. The only major concern is the writing that require a careful revision to make the article much more accessible and readable. The author have the tendency to write in an non-informative way that leads to many meaningless paragraphs that are not very useful, maintaining the reader without clear information. For example, at the end of the introduction, the last paragraph said that you have "counter balancing contribution to adhesion and collective migration", or "differential localization and antagonistic effect on cell behavior". This type of writing while pointing to a difference leave the reader without a clue on the message and what each molecule is contributing to. This is present in many places in the article. The entire start of the discussion in particular has the same problem. We have "opposite effect" or "diametrically opposed phenotype" but no clue on who is doing what and where. All the "suspense paragraph" or unclear transitions should be also eliminated. This is very important for the clarity. Readers can have different level of attention and comprehension and repeating much more directly the take home message and the results will help tremendously for the impact of this nice work.

We thank the reviewer for this criticism, we have made extensive changes in the manuscript, including rewritten large parts of the discussion. We have made sure to clearly explicit each conclusion/statement. We have systematically removed superfluous transitions.

Referee #2:

Review for Aslemarz et al "An EpCAM/Trop2 mechanostat differentially regulates collective behaviour of human carcinoma cells"

Summary

In this manuscript, Aslemarz et al investigate how single or double knock-down of EpCAM and its paralog Trop2 impacts cell and tissue wetting across scales. First, knock-downs are performed in MCF7 cells, and the effective spreading area and solidity (circularity) of spheroids and single cells are determined. Through this, the authors find that EpCAM KD causes increased spreading ("wetting"), while Trop2 KD causes reduced spreading. Next, the authors investigate how KD impacts phospho-myosin2 status, observing that EpCAM KD leads to increased peripheral pMLC, while Trop2 leads to increased pMLC at inner junctions. The authors then examine Vinculin accumulations at cell-substrate and cell-cell adhesions, observing that Trop2 KD appears to increase relative vinculin levels at both adhesion types. The authors then connect this to measurements of force generation at cell-substrate adhesions using traction force microscopy, finding that both EpCAM and/or Trop2 KD lead to increased contractile energy/stress magnitude. They then infer relative cortical tension values and adhesiveness in each KD condition by measuring contact angles between heterotypic cell doublets. Using these experimental data as inputs, the authors then use a

cellular Potts model to perform simulations of cellular aggregate spreading events in the different KD conditions, which match the experimentally observed phenotypes. The authors then investigate asymmetries in EpCAM and Trop2 localization patterns across different cellular interfaces, finding that EpCAM has a higher dorsal/ventral membrane localization, while Trop2 has higher relative localization at inner cell-cell junctions. Finally, the authors perform EpCAM and Trop2 KD in two additional cell lines and find consistent results in terms of spreading behavior in single cells and spheroids.

Overall, the manuscript does a nice job of characterizing the phenotypes between EpCAM and Trop2 KDs. The overall finding that knock-down of these two paralogs is quite surprising, and the data presented provides a nice explanation of why this is the case. The authors generally take a quantitative approach to describe the cellular and tissue-level behaviors and using mostly qualitative arguments attribute these changes in wetting behavior to changes in active tensions at cellular interfaces. This is also nicely complemented with simulations using a cellular Potts model that mimics the experimental observations. There are still a number of aspects of this manuscript that could be improved prior to acceptance for publication, which are listed in a point-by-point manner below:

Major Points

1. It is not clear whether Box 1 is really necessary in the main text. This theory, mostly based on the Young-Dupré equation, has been described in many previous publications applying this to biological systems, albeit with slightly different notation (e.g. Winklbauer, 2015; Maître et al, 2012; Turlier and Maître, 2015). Sometimes this is not always clear from the text, and more effort could be given to discussing this in the context of previous publications, especially in developmental biology, where this theory has been applied extensively, and also in more recent publications using in vitro systems of cell aggregates undergoing active wetting on deformable surfaces (e.g. Pérez-González et al, 2019, Nat. Phys.; Yousafzai et al, 2022, PRX). Also, the theory is mostly used to make qualitative arguments throughout the paper, as tensions are not measured directly. While I do believe that this framework allows for a better understanding of the phenotypes, as it is not used to make quantitative measurements for much of the paper, I am not sure how much focus should be placed on the theory in this case.

We have now moved the box out of the main manuscript figures, and placed it as figure of the “expanded version”. This will make this theory easily available, if needed, for the general readership (beyond of the biophysics and morphogenesis community).

Yes, obviously the theory has been already described in many publications. We hope that in the revised version, it is now unequivocal that this theory has been already extensively applied to embryonic models, as well as to cell aggregates.

We have also tried to make it clear that, although most data were only qualitatively related to the balance of tension, we did evaluate actual relative tensions and adhesiveness based on geometry of cell doublets. Furthermore, to set the values for the CPM model, we have largely relied on the geometry of single cells and doublets, thus on the balance of tensions.

We agree that we needed to discuss more about other publications, especially those reporting properties of aggregates on soft substrates. Discussing Pérez-González et al, 2019 was particularly useful, as it allowed to highlight both shared aspects and distinct characteristics of this EpCAM/Trop2-dependent system.

2. As EpCAM and Trop2 are paralogs and have potentially overlapping functions, it is possible that there are some regulatory mechanisms to compensate for expression levels. The authors also address this in the Discussion section. It would therefore be important and should be possible to show that expression levels are independent. Ideally this could be done by Western Blot.

This was an obvious point that we had to address. We have added in the supplementary material (Appendix Fig. S4) quantification of Western Blots. We find that EpCAM is slightly upregulated in Trop2 KD, while Trop2 does not seem impacted by EpCAM KD.

3. In Fig. 2, it is shown that Trop2 KD leads to increased pMLC accumulation at internal cell-cell junctions compared to at the cluster edge. In a previous publication, reduction of pMLC at the internal junctions of similarly sized cell clusters on collagen networks was shown to be regulated by DDR1, Par3/6, RhoE, and ROCK (Hidalgo-Carcedo et al, 2011, NCB). It would be insightful to determine whether Trop2 is associated with any of these components and to discuss the results in the context of this previous publication.

We would be keen to understand the molecular determinants that differentiate EpCAM from Trop2 and looking into the DDR1-dependent complex is indeed one of the potential avenues. However, this would be a whole new project, beyond the scope of this present study, and for which at the moment we do not have the resources nor workforce. We mention the reference to DDR1-Par complex in the discussion.

4. One of the most striking changes following Trop2 KD is the upregulation of Vinculin at cell-cell junctions. As the effects of Trop2 KD depend on changes in myosin contractility and cell-cell adhesion, increased vinculin recruitment to junctions seems to be an important observation and could be indicative of increased tension at junctions (le Duc et al 2010, JCB; Leckband and de Rooij, 2014 [already cited in manuscript]). Although this seems to be more striking than the difference in vinculin accumulation at FAs, this is never really followed up. Instead, the focus is directed toward stresses at cell-substrate adhesions. Additional investigation of mechanosensing at cell-cell junctions would help to understand this mechanism.

We agree that we may not have put enough emphasis on the cell-cell contact pool of vinculin. To get some more insight, we have now performed double staining for β -catenin and vinculin. Both EpCAM and Trop2 KD did increase β -catenin at contacts, just as E-cadherin in our previous experiments. Quite strikingly, however, the relative vinculin/ β -catenin ratio at cell-cell contacts, identified through the β -catenin signal changed in opposite ways, lower for EpCAM KD, higher for Trop2 KD. Furthermore, we made an interesting observation, which we mention in the manuscript: In wild type and EpCAM KD, while both β -catenin and vinculin accumulate at contacts, they do show a very clear complementary pattern: Vinculin is weak at β -catenin-rich contacts, and vice versa. This can be interpreted considering that the well-known property of the cadherin-catenin complex to repress local cortical actomyosin contractility. In Trop2 KD, however, high β -catenin and vinculin often colocalize. We consider this as additional indication that in Trop2-depleted cells, tension gets more acute at adhesive interfaces, shifting them from a “soft” adherent state to a more tensile state. The situation is thus quite similar as for cell-matrix contacts: Cells are still able to make strong adhesion interactions, but the increased contractility of the cortex makes cells and tissue stiff, which negatively impacts on both spreading and tissue cohesion.

5. The Vinculin stainings and TFM data seems somewhat inconsistent. On one hand, vinculin accumulation at FAs seems to be highest for the Trop2 KD. On the other hand, total traction energy and traction stresses seem to be highest for EpCAM KD, where no change in Vinculin at FAs was observed. As FA assembly is mechanosensitive and higher tractions are typically associated with more mature, FAs, how can this difference be reconciled? And what is meant by the following "adhesion reinforcement in Trop2 KD cells was less efficient" (p. 9)?

We agree that the term adhesion "reinforcement" is ambiguous, and we now avoid it throughout the manuscript.

About vinculin staining and TFM data: In fact, "total contractile energy" (E_c) from TFM should be compared to "total vinculin" (=integrated intensity in Figure 3A'), since both reflect the global stress that cells exert on the matrix.

In contrast, Figure 3A" reports vinculin/area values, which we can call vinculin "density". The latter is related to the balance of tensions, which is not directly related to TFM-measured traction stresses. Vinculin intensity in large cell groups (Figure 3B') was measured as "average intensity in a field, which also reflects "density". This should be compared to vinc/area in Fig.3A", but not to TFM values (Fig.4D). For these cell groups, we did not calculate an integrated intensity, because it was not possible to unambiguously allocate FAs to a particular cell.

We now explicitly make the distinction between these two types of values in the text (see also next point).

6. The TFM data is mostly used to argue that pMLC activity and contractility are higher for both KD conditions. However, the actomyosin structures on the "ventral" surface where the FAs are located is typically very different from the actomyosin cortex on the "dorsal" surface. This should be treated more carefully in the text, as it is not clear how increased tension on FAs relates to cortical tension (γ_m). Also, one is left wondering whether any information related to the cell-matrix tension can be deduced from the TFM experiments. Do the measured traction stresses/energy relate to interfacial tensions in any direct way? It would be helpful to directly address this in the text.

Indeed, the major direct point of TFM was the demonstration that contractility was higher in both KD. Another important result was the increased force exerted on the cell-cell interface. These remain key evidence for this study.

However, force and energy values extracted from this assay cannot be directly related to interface tensions. This is indeed an important point, we have now made it clear in the text.

Furthermore, we have now also emphasized that H patterns impose a constraint configuration to the cells. These constraints are very useful to compare mechanical properties under similar conditions, but cells adopt very different shape than when free to spread and migrate. In particular, all cells spread to a similar extent, including for siTrop2 KD, which otherwise makes cells round and tall. Thus "wetting" (-> interfacial tensions) cannot be compared to "free" cells.

This being said, TFM data are a source of indirect information, provided some assumptions. The flat morphology implies that γ_m is oriented almost horizontally, thus close to the plane of traction on the matrix. Considering that the KDs increased both total contractile energy and the adhesive force F_{cc} , one may make the reasonable assumption that a significant part of the tension is transmitted via the lateral/dorsal free edges. From there, we can consider that

ym must be indeed related to contractile energy. This is clearly an extrapolation, but this assumed relationship is consistent with the rest of our data.

The reviewer is also raising another important point about the contribution of stress fibres. In fact, the coarse grain model of interfacial tensions used in our study does not require to discriminate between the forces exerted by various subcellular structures, here the cell cortex and stress fibres. These interfacial tensions are simple terms that integrate all multiple inputs, which positively or negatively impact on contractility and on adhesion. We have now explicitly explained this aspect in the text. Note that for cells/tissues on soft substrate, “classical” stress fibres can hardly be detected, they must be mostly “merged” with the cell cortex, similar to the links for cadherin cell-cell adhesion.

7. Several times throughout the manuscript, the authors reference expression data from public repositories (e.g. Expression Atlas, p. 4, 16; Protein Atlas, p. 12, 16). If this expression data is available, would it be possible to simply generate some plots based on this meta-analysis? This would be more informative for readers than simply citing the repositories.

We fully agree. We have prepared a supplementary figure (Appendix Fig S1), which presents values extracted from the data base for selected cell lines, cancers and normal tissues.

Minor points

1. The contact angle at the edge of the spreading spheroids is important for understanding spreading mechanics, and the authors include this in Fig. 5. Could this also be measured in the other figures (e.g. Fig. 1H', Fig. 6A",B",C")? It is also unclear why the authors chose to bin the contact angles and define a new "round/spread" scale in Fig. 7C. Although this scale is described well in Fig. S5C, it seems slightly arbitrary and would probably be more straightforward to measure an average (mean or median) contact angle.

Yes, we agree. Following the reviewer's comment, we now provide quantification for spheroid morphology, i.e. height/diameter ratio and measures of contact angles (Figure 1J,J'), which indeed reflect spreading/wetting.

We also show estimates of contact angles for single MCF7 cells, in the new Appendix Fig. S5. However, these angle values were very approximative, due to the highly irregular shapes of the cells and the impossibility to find objective criteria to define actual cell “edges” and “dorsal slopes”. Height/diameter ratios were much more robust measurements (Figure 1K”), which is the reason why we used the latter in the simulations (Figure 5I and Appendix Section 1). We have however verified that these simulated shapes had edge angles reasonably close to the measured angles.

It was even more difficult to properly measure an “average” angle for HCC1500 cells, and this had been the reason to go for a score based on “categories”. For the revision, we tried once more to estimate these angles, but we felt that the values obtained were more arbitrary than our “binned” categories, thus we are reluctant to include them. Here below are examples of frequent shapes for which determining a “contact” angle is far from obvious.

2. It is unclear what is presented in Fig. 1G. Are these outlines from actual data, or simply cartoons? I do not think these panels are necessary. Maybe this idea could be better presented by simply outlining the cell aggregates in H/H' for example with a dotted line.

We had drawn the outlines in separate panels to avoid masking the details of the IF images. We have improved these outlines to fit more precisely the actual data, and clarified it in legend. We have slightly reorganized the figure, switching order of these specific panels.

3. The difference in height is striking between conditions presented in Fig. 1 H'. Could this be quantified in addition to the area increase and solidity? Even better would be to measure contact angles.

Yes, we have added both height/diameter and contact angles (see also below).

4. In Fig. 1I', the normalization is slightly unconventional. Height/Area would give a unit of 1/length (e.g. μm^{-1}). No unit is given on the plot. Perhaps simply an aspect ratio from the xz projection (which would be unit-less) would be better here and/or the contact angle at the cell edge.

This was indeed a mistake, which is now corrected. We present height/diameter, the latter being an "average diameter" calculated from the cell cross-section area, in order to normalize for highly variable irregular cell shapes. This is also specified in Material and Methods.

5. In the quantifications of the dorsal-ventral localizations of EpCAM and Trop2 (Fig. 6E, S6C), there is a lower D-V gradient for Trop2 in both MCF7 cells and HCC1500 cells, which the authors point out in the text. However, the gradient is in the opposite direction for the two cell lines. For MCF7 cells, the ratio is >1 , meaning that both EpCAM and Trop2 are enriched on the dorsal side, while for HCC1500 cells, the ratio is <1 , meaning that EpCAM and Trop2 are enriched on the ventral side. This could be made clearer in the text, and it would be helpful to discuss the implications for the interfacial tensions.

This very good point is now brought in results and discussion. The same shift is also seen for the dorsal (free edge)/ contact ratio. In principle, these different D/V ratios could imply that in MCF7 cells both EpCAM and Trop2 preferentially influence tension at free edges (γ_m), in HCC1500 more tension at matrix and cell-cell contacts. The fact that HCC1500 cells are clearly spreading less than MCF7 cells is here interesting. We speculate that both EpCAM and Trop2 may both accumulate at sites of highest tension, which differs between cell types (the free edge of HCC1500 cells). But it is hard to make sensible predictions on the actual balance of tensions, as other factors may impact on these tensions in very different ways in each cell line. Yet, it appears that the relative bias of Trop2 toward contact interfaces compared to EpCAM is still sufficient to shift the balance of tensions the same way in different cells.

Figures: Correspondence between first submission and revised manuscript

Main figures

Fig. 1 remains Fig.1. Added morphology measurements for spheroids (J,J').

Fig. 2. Unchanged except for a shift in order of graphs J-L.

Fig. 3,4. Unchanged.

Fig. 5. Added shapes of single cells and doublets used to set simulation energies.

Fig. 6. Unchanged.

Fig. 7. Split in new figures 7 (HCC1500 cells), 8 (MDA-MD-453 cells), 9 (summary).

New figure 7 includes data previously in Fig. S5 (spheroids) and S6 (EpCAM and Trop2 IF).

New figure 8 includes data previously in Fig. S7 (complete data on spreading and morphologies, EpCAM and Trop2 IF).

Expanded view:

Previous Box 1 is now Fig. EV1.

Fig. EV2 includes previous spheroid IF from Fig. S2, complemented with new phalloidin stainings of small spheroids.

Fig. EV3 New figure presenting new data on beta-catenin – vinculin double staining of cell groups.

Fig. EV4 New figure presenting images of late spheroid spreading and corresponding simulation.

Fig. EV5 Previously Fig. S5 (HCC1500 morphologies). EpCAM-Trop2 IF moved to main Fig. 7.

Appendix

Appendix includes all remaining supplementary Figures, with the following additions:

Appendix Figure S1: Comparison expression levels of EpCAM and Trop2 in various cell lines and types, assembled based on public database

Appendix Figure S4: New data. Western blots for total EpCAM and Trop2.

Appendix Figure S5: New data. Estimated contact angles for single MCF7 cells.

Appendix Figure S6: New data. Quantification of phalloidin for cell groups, see main Fig. 2.

Dear Francois,

Thank you for submitting the revised version of your manuscript. We have now received input from one of the original reviewers, who finds that their previous concerns have been addressed satisfactorily and now recommends publication of the study. Therefore, there now remain only a few editorial points that need addressing before I can extend official acceptance of the manuscript:

1. Please check if the email addresses provided for the authors Artur Ruppel (artur.ruppel@univ-grenoble-alpes.fr) and Christine Fagotto-Kaufmann (Christine.FagottoKaufmann@crbm.cnrs.fr) are correct, as the emails sent to them were returned.
2. Please submit up to five keywords.
3. Please upload the main and EV figures as individual production quality figure files in the .eps, .tif, or .jpg format (one file per figure).
4. In the Author Checklist file, please indicate in the column E the manuscript section in which the information can be found.
5. Figure panels 1F, 6B-C, 6J, 7D are not mentioned in the manuscript text. Please add the appropriate callouts.
6. Please add a "Disclosure and competing interests statement" section. Further info: <https://www.embopress.org/page/journal/14602075/authorguide#conflictsofinterest>.
7. We require a Data Availability Section at the end of Materials and Methods. As far as I can see, no data deposition in external databases is needed for this paper. If I am correct, then please state in this section: This study includes no data deposited in external repositories. Further information can be found at <https://www.embopress.org/page/journal/14602075/authorguide#dataavailability>
8. Please remove movie legends from the manuscript text file and zip as README files together with each movie file. Further information is available here: <https://www.embopress.org/page/journal/14602075/authorguide#expandedview>
9. Please check the numbering of Appendix Figures - there are two Appendix Figures S2, and the title and the legend do not match for Appendix Figures S2 and S3. Please also remove the list of Appendix Figures from the manuscript text file.
10. In our standard image integrity check, we noted a reuse of the image panels between Figure 7A & B and Figure EV5 A, A' & B. If this is intentional, please indicate in the figure legends that the images are derived from the same experiment/sample.
11. Our data editors have flagged the following issues in figure legends that need correcting:
 - Please note that the sub-figures 7F'-F', G-G', H-H' are mislabeled as figure 7A', B, B', C, C' respectively in the manuscript. Please check and correct.
 - Please note that the sub-figure legend for fig 8c-c are mislabeled as figure 8C in the manuscript. Please check and correct.
 - Please add the legend for figure panels 7E and EV2 D'.
 - Please provide the exact p values in the legends of figures 1E, F, J', K', K'; 2H-J, L; 3A', A', B', B'; 7H, H', 8C, C', E; EV2 B', D', F, H; EV5 D.
 - Please indicate the statistical test used for data analysis in the legends of figures 7E'.
 - Please define the box plots in terms of minima, maxima, centre, bounds of box and whiskers, and percentile in the legends of figures 1F, J, J', K, K'; 2C-F, H-L; 3A', A', B', B'; 4D-G, K, L; 6E-G, I, I'; 7H, H'; 8C', C', E; EV2 B', D', E, E', F, G, H, I; EV3 D-F.
 - Please note provide information on the number and nature of replicates in the legends of figures 1J'; 4E, F, G.
 - Please describe the nature (e.g., biological or technical) of replicates in the legends of figures 1F, J, K', K'; 2C-F, H-L; 5I', 6E-G; 7E'; EV3 D-F.
 - Please define the error bars in the legends of figures 5I', 7H'.
 - Please note that the scale bar is missing for figure 4I.
 - Please define the scale bar for figures 7E, E', E'.
 - Please note that scale bar and its definition are missing for figure EV2 D'.
 - Please define the letter "B" in the legend of figure 4I.
 - Please define the yellow arrow heads and yellow concave arrow heads in the legend of figure 4I.
 - Please define the blue dashed box, red dashed line in the legend of figure 8B', D.
 - Please define the blue arrow heads in the legend of figure EV4C'.
 - Please define the white arrows, white arrow heads, blue arrows, blue arrow heads, white concave arrows in the legend of figure EV5 B.
12. Please submit source data files as requested by our data editor.
13. Papers published in The EMBO Journal are accompanied online by a 'Synopsis' to enhance discoverability of the manuscript. It consists of A) a short (1-2 sentences) summary of the findings and their significance, B) 3-4 bullet points highlighting key results and C) a synopsis image that is 550x300-600 pixels large (width x height, jpeg or png format). You can either show a model or key data in the synopsis image. Please note that the image size is rather small and that text needs to be readable at the final size. Please send us this information together with the revised manuscript.

With best wishes,

Ieva

We realize that it is difficult to revise to a specific deadline. In the interest of protecting the conceptual advance provided by the work, we recommend a revision within 3 months (31st Dec 2024). Please discuss the revision progress ahead of this time with the editor if you require more time to complete the revisions.

Referee #2:

In the revised manuscript, Aslemar et al performed several new experiments and added a number of new quantifications to the existing data. The authors also made substantial changes to the manuscript text. I feel that the revised manuscript is much improved. The new experiments and analysis help to substantiate the authors' conclusions, and overall, the manuscript is more clearly presented. In particular, the new data and analysis on Vinculin and β -catenin are very nice and provide a deeper understanding of the overall message of the paper. In summary, the authors have done a nice job addressing all of the reviewers' concerns, and I am happy to recommend this manuscript for publication for the wider readership of the EMBO Journal.

The authors addressed the minor formatting issues.

Dear Francois,

Thank you for addressing most of the remaining editorial issues. I apologise for the delay in the processing of your manuscript due to the high number of pre-acceptance submissions that we receive at the moment. Upon checking your manuscript, I am afraid that I noticed a few remaining issues that need to be fixed before I can accept it for publication:

1. In our standard source data check we noticed a couple of instances of numerical duplications that I wanted to check with you on. I have attached the relevant files with the numerical repetitions labelled in colour. For figure panels 1K, you will notice in exp 4 and exp 5 blocks of three repeating values. For figure 2, we have marked in colour repetitive numerical duplications that are present in all figure panels. In the source data for figure 6l, there is a block of two repeating values in exp 4. Please check and correct if needed. A brief explanation would be very helpful.

2. I would like to propose minor textual edits in the synopsis and abstract. I have also written a short blurb that will accompany the title of your manuscript in our online table of contents. Please take a look at the text below and let me know if any edits or corrections are needed.

Abstract:

In the third sentence: "Here, we investigate the morphogenetic impact of the high EpCAM and Trop2 levels typically found in epithelial breast cancer cells, using spheroids of MCF7 cells as an in vitro model."

Blurb:

Two closely related cell surface proteins affect cortical tension in different domains and differentially regulate cell cluster cohesion and spreading.

Synopsis:

Cell adhesion and cell cortex contractility are important determinants of the morphogenetic properties of solid tissues, including tumours. This study shows that EpCAM and Trop2, two closely related surface markers of carcinoma, play antagonistic roles, respectively opposing or favouring cohesion and collective migration of breast cancer cells.

- Both EpCAM or Trop2 repress myosin activity and cortical contractility.
- EpCAM tends to act preferentially on the cell cortex at free edges, Trop2 on matrix and cell adhesive contacts, subtly controlling the balance of tensions at interfaces.
- Consistently, the two molecules show differential enrichments at the corresponding cell cortices.
- MCF7 spheroids depleted of EpCAM extensively spread while remaining compact, while those depleted of Trop2 show lower cohesion and spreading than wild type spheroids.
- Effects of EpCAM and Trop2 can be accurately modelled based on the biophysical analogy with the phenomenon of wetting-dewetting.

Thank you again for giving us the chance to consider your manuscript for The EMBO Journal. I look forward to receiving your input on these final points.

With best wishes,

leva

leva Gailite, PhD
Senior Scientific Editor
The EMBO Journal
Meyerhofstrasse 1
D-69117 Heidelberg
Tel: +4962218891309
i.gailite@embojournal.org

We realize that it is difficult to revise to a specific deadline. In the interest of protecting the conceptual advance provided by the

work, we recommend a revision within 3 months (21st Jan 2025). Please discuss the revision progress ahead of this time with the editor if you require more time to complete the revisions.

The authors addressed the remaining editorial issues.

Dear Francois,

Thank you for addressing the final editorial points. I am now happy to inform you that your manuscript has been accepted for publication in the EMBO Journal.

We will wait to hear from you regarding the source data update, but otherwise we should have everything together for transferring your manuscript to our publishers.

If you have any questions, please do not hesitate to contact the Editorial Office. Thank you for this contribution to The EMBO Journal and congratulations on a nice study!

Best wishes,

leva

leva Gailite, PhD
Senior Scientific Editor
The EMBO Journal
Meyerohofstrasse 1
D-69117 Heidelberg
Tel: +4962218891309
i.gailite@embojournal.org
